# Detecting Perspective Shifts in Multi-Agent Systems

Eric Bridgeford [1]    Hayden Helm [1]

## Abstract

Generative models augmented with external tools and update mechanisms (or *agents*) have demonstrated capabilities beyond intelligent prompting of base models. As agent use proliferates, dynamic multi-agent systems have naturally emerged. Recent work has investigated the theoretical and empirical properties of low-dimensional representations of agents based on query responses at a single time point. This paper introduces the Temporal Data Kernel Perspective Space (TDKPS), which jointly embeds agents across time, and proposes several novel hypothesis tests for detecting behavioral change at the agent- and group-level in black-box multi-agent systems. We characterize the empirical properties of our proposed tests, including their sensitivity to key hyperparameters, in simulations motivated by a multi-agent system of evolving digital personas. Finally, we demonstrate via natural experiment that our proposed tests detect changes that correlate sensitively, specifically, and significantly with a real exogenous event. TDKPS is the first principled framework for monitoring behavioral dynamics in black-box multi-agent systems – a critical capability as generative agent deployment continues to scale.

## 1. Introduction

The general improvement of Large Language Models (LLMs) has spurred the development of generative systems that use tools to interact directly with their environment (Yee et al., 2024). Consider a system consisting of a web-crawler with access to the internet, a database for storage, and an LLM. When prompted, the system begins to scrape the internet based on the prompt. The LLM assesses the relevance of each datum before it is added to the database. Once the scrape is complete, the LLM provides a response based on the data stored in the database. We refer to a system whose behavior can be affected by a change in its tools (e.g., the web-crawler & storage), a change in its base model (e.g., the LLM), or a change in its environment (e.g., the internet), broadly, as a (generative) "agent."

Recent progress in the design of generative agents has led to their deployment in increasingly complex environments, often populated by other agents performing similar information-gathering and reasoning tasks. These environments are characterized as multi-agent systems in which agents interact, exchange information, or otherwise influence each other's behaviors. Multi-agent systems are inherently complex (Han et al., 2024): each agent typically consists of multiple dependent components; their update mechanisms are often loosely defined and depend on interaction with their environment; and the effects of one agent's actions on itself and others can be convoluted. Given the rise in popularity of generative agents for different tasks, developing methods for understanding the dynamics of multi-agent systems is critical to advancing the reliability and safety of agent use in shared environments.

One of the most fundamental statistical challenges in studying multi-agent systems is determining if, or when, an agent's (or collection of agents') behavior(s) have changed. In this paper, we address this problem in the black-box setting, where an agent's internal mechanisms are inaccessible and only its inputs and outputs are available for analysis. This setting is the most universal and realistic regime for monitoring modern multi-agent systems, given that an agent may be proprietary, may have access to private external tools, or may have access to privileged information.

**Contribution.** The framework developed herein – the Temporal Data Kernel Perspective Space (TDKPS) – is the first to enable principled statistical inference on general agent dynamics in black-box multi-agent systems. We describe and characterize the statistical properties of the first tests for temporal change detection at the agent- and group-level.

---

[1]Helivan, San Francisco, CA. Code, data, and replication scripts are available at https://github.com/helivan-research/maps. Correspondence to: Hayden Helm <hayden@helivan.io>.

*Proceedings of the 43rd International Conference on Machine Learning*, Seoul, South Korea. PMLR 306, 2026. Copyright 2026 by the author(s).

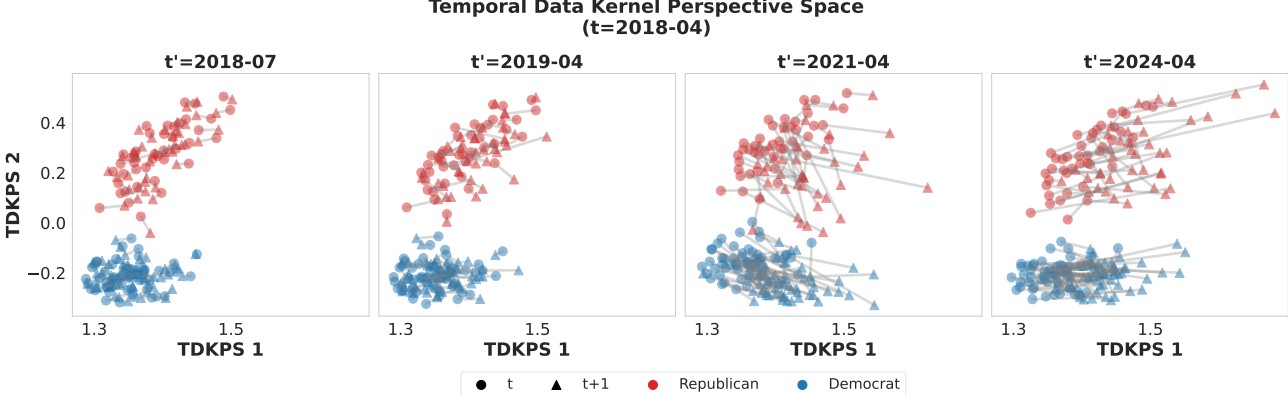

*Figure 1.* The $T = 2$, 2-d Temporal Data Kernel Perspective Space ("TDKPS") of a multi-agent system consisting of generative agents parameterized by different, dynamic retrieval datasets. Each dot/triangle is an agent. TDKPS enables interpretable and principled analysis of multi-agent systems in the black-box setting. For more experimental details, see Section 3.1.

## 1.1. Related work

**Multi-agent systems**   The majority of research on multi-agent generative systems has been in the context of computational sociology and behavioral simulation (Park et al., 2023; McGuinness et al., 2025; Chen et al., 2025a). Broadly, prior work simulates predefined agent architectures interacting within sandboxed environments (Park et al., 2023; Piao et al., 2025) or along fixed interaction graphs (Papachristou & Yuan, 2024; Helm et al., 2024; Chuang et al., 2024). Quantitative tracking of agent dynamics in these studies is typically limited to aggregate system-level statistics of group behavior (Sun et al., 2025; Chen et al., 2025b; Tran et al., 2025). The growing deployment of agentic systems in live, open environments motivates our more general treatment of black-box multi-agent systems and our approach to characterizing general agent behavior.

**Representations of models**   A core component of any generative agent is the LLM that drives its reasoning, tool use, and responses. Understanding differences between models is therefore a natural starting point for understanding differences between agents. Numerous methods embed language models into low-dimensional spaces—via their internal representations (Duderstadt et al., 2024; Huh et al., 2024), parameter weights (Putterman et al., 2024), or responses to shared queries (Acharyya et al., 2025)—enabling standard statistical inference. Among these, the Data Kernel Perspective Space (DKPS) (Helm et al., 2024; Acharyya et al., 2025; Helm et al., 2025a) is most directly relevant: it represents black-box models via response similarities to a reference query set. To our knowledge, no existing framework provides representations for studying the dynamics of general agent behavior.

**Inference on structured objects**   Detecting temporal changes in multi-agent systems requires statistical methods for structured, high-dimensional data. Prior work on structured objects—latent position graphs (Tang et al., 2013), connectomes (Chung et al., 2021; Bridgeford et al., 2025), physiological signals (Chen et al., 2022)—typically constructs null distributions via permutation tests (Székely & Rizzo, 2004; Gretton et al., 2012). The most closely related is (Tang et al., 2017), which defines relationships via network edges rather than behavioral distributions. Critically, existing methods assume either univariate data, unpaired observations, or employ parametric assumptions for inference. The assumptions typically stem from the difficulty of obtaining multiple realizations from high-dimensional structured objects across time. The novel non-parametric tests we propose and study are tailored specifically to the black-box multi-agent system setting.

## 2. The Temporal Data Kernel Perspective Space

In the black-box setting, a generative model $f : \mathcal{Q} \to \mathcal{X}$ is a random mapping from a query space $\mathcal{Q}$ to a response space $\mathcal{X}$. Given a query, the query response $f(q)$ is sampled from $P(q)$ and the query responses $f(q)_1, \ldots, f(q)_R$ are *i.i.d.* samples from $P(q)$. We let $g : \mathcal{X} \to \mathbb{R}^p$ be an embedding function that maps a query response to a real vector. The embedded query responses $g(f(q)_1), \ldots, g(f(q)_R)$ are *i.i.d.* samples from $P_{g(f)}(q)$. For practical purposes we deal exclusively with the embedded query responses. *In this paper, we use the terms "model" and "agent" interchangeably: both refer to random mappings from a query space to a response space.* We sometimes refer to agents as models (and vice versa) when the context is clear.

We observe $N$ agents $F^{(t)} = \{f_1^{(t)}, \ldots, f_N^{(t)}\}$ over $T$ timepoints. For queries $Q = \{q_1, \ldots, q_M\}$, we let $X_n^{(t)} \in \mathbb{R}^{M \times R \times p}$ denote the tensor containing the embedded query

responses $f_n^{(t)}(q_m)_r$ for agent $n$ at time $t$. We further let $\bar{X}_n^{(t)} \in \mathbb{R}^{M \times p}$ denote the matrix whose $m$th row is the average embedded query response from the $n$th agent at time $t$ to the $m$th query, $\bar{X}_{nm\cdot}^{(t)} := \frac{1}{R} \sum_{r=1}^{R} g\left(f_n^{(t)}(q_m)_r\right)$. We define $D^{(t,t')}$ to be the $N \times N$ pairwise distance matrix with entries

$$D_{nn'}^{(t,t')} := \left\| \bar{X}_n^{(t)} - \bar{X}_{n'}^{(t')} \right\|_F.$$

Let $\tilde{D}$ be the $T \cdot N \times T \cdot N$ distance matrix with block entries equal to $D^{(t,t')}$ for $t, t' \in \{1, \ldots, T\}$.

The $d$-dimensional Temporal Data Kernel Perspective Space (TDKPS) representations of the agents are the low-dimensional vectors $\psi_n^{(t)}$ that are a solution to

$$\left(\psi_n^{(t)} : n \in [N]\right)_{t \in [T]}$$
$$= \operatorname{argmin}_{z_i \in \mathbb{R}^d} \sum_{i,j=1}^{T \cdot N} \left(\|z_i - z_j\| - \tilde{D}_{ij}\right)^2. \quad (1)$$

The TDKPS enables treating the otherwise-complex analysis of multi-agent systems as a dynamic process in a low-dimensional Euclidean space. Analysis using TDKPS is dependent on the query set $Q$ and the embedding function $g$. When $T = 1$, TDKPS reduces to the non-dynamic Data Kernel Perspective Space framework first introduced in (Helm et al., 2024) and analyzed in (Acharyya et al., 2025; Helm et al., 2025a). For algorithmic simplicity, we assume $\tilde{D}$ is a Euclidean distance matrix. Therefore, the solution to Eq. (1) is found via classical multidimensional scaling of $\tilde{D}$ (Torgerson, 1958). As an example, the TDKPS representations of a collection of agents for various pairs of time points (e.g., $T = 2$) are shown in Figure 1. Each dot / triangle is an agent. The data is described in Section 3.1.

# 3. Detecting Perspective Shifts

We first introduce one of our motivating multi-agent systems and a corresponding simulation data generation process. After describing the data, we propose hypothesis tests to detect agent- and group-level changes, characterize their Type-I error rates and powers in simulation, and apply them to our motivating multi-agent system.

## 3.1. Data

**System of Digital Congresspersons**  One of our motivating applications is a system of evolving digital congresspersons introduced in (Helm et al., 2025b). Specifically, we consider a collection of Ministral-8B-Instruct-2410 agents (Mistral AI Team, 2024) each with access to a different Congressperson-specific database containing all of their Tweets while in office. We consider only Congresspersons that were in office continuously from 2016 to 2025 and that

had more than 100 Tweets by the end of the first quarter of 2018. This left us with $N = 99$ digital congresspersons.

We consider three different query sets in our real data analysis – 100 questions related to public health, 100 questions related to general politics, and 100 questions related to candy & chocolate. The set of questions from all three topics were generated by ChatGPT (OpenAI, 2025). The prompts used to generate the query sets are provided in Appendix A.1.

For each question the two most relevant[1] Tweets were retrieved for each Congressperson up to time $t$ and put into the following prompt structure:

System prompt:

You are U.S. Congressperson {name}. You must answer all questions in the first person as if you are them. Do not answer with any lists and do not use any hashtags. Answer as concisely as possible.

User prompt:

Here are some of your Tweets related to your opinion on {topic}: {concatenated_tweet_string}.\n\n{question}

For each completed prompt structure, Ministral-8B-Instruct-2410 was queried $R = 25$ times at the end of the first and third quarter of each calendar year from 2018 to 2024. Finally, each response was embedded into $p = 768$ dimensions using nomic-embed-v1.5 (Nussbaum et al., 2025), yielding the data tensor $X \in \mathbb{R}^{14 \times 99 \times 100 \times 25 \times 768}$ for each topic.

Continuing our simple example agent from the introduction (LLM, web-crawler, internet, and database), each digital congressperson can be thought of as an instance of this agent that, when prompted with a query, initiates a scrape of the official Twitter account associated with a Congressperson from the time of prompting to a pre-determined cut-off, stores the Tweets, and uses only the top two most relevant Tweets for context when producing a response. The TDKPS of the agents induced by the public health questions for four different pairs of timepoints are shown in Figure 1.[2]

We emphasize that the dynamic process that generates each congressperson's tweet history innately involves congresspersons interacting with one another. For example, they respond to each other publicly, co-sponsor and debate shared legislation, react to the same news events, and participate in overlapping political coalitions. While this interaction graph is complicated and latent (we do not observe it directly), the observed data (the tweets that populate each agent's retrieval database) is dependent on it. This setting, where the operator does not have access to the interaction

---

[1]Relevance determined by the cosine similarity between embedded Tweets and the embedded question.

[2]Generating 25 responses from all agents at a given $t$ for a given topic took approx. 3 hours on a single H100. The generation of the data took approx. $3 \cdot 14 \cdot 3 = 126$ H100 GPU hours.

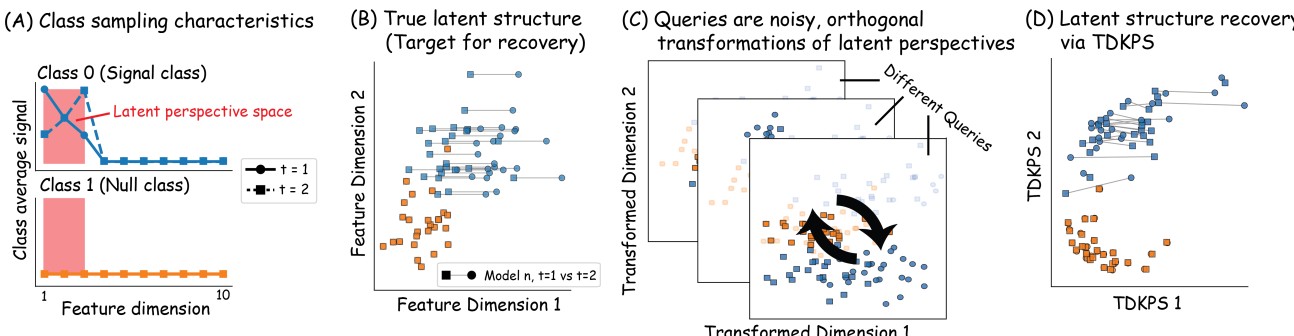

*Figure 2.* **Simulation design and TDKPS schematic.** Illustration with $N = 50$ agents, $p = 10$ total dimensions, $p_s = 3$ signal dimensions, and effect size $\tau = 1$. **(A)** Class-specific temporal dynamics: Class 0 (blue) exhibits temporal change transitioning from front-loaded ($t = 1$) to back-loaded ($t = 2$). Class 1 (orange) shows no temporal change. **(B)** True latent structure in the first two dimensions, with lines connecting each agent across timepoints. **(C)** Observed query responses result from random orthogonal transformations of the latent space (three example queries shown) with added measurement noise, obscuring the latent structure. Illustrated is a random rotation, which is a type of orthogonal transformation. **(D)** TDKPS embedding recovers the relational structure from (B).

graph or internal states of the underlying dynamics and can only observe agents' input-output behavior, is precisely the black-box regime that motivates our framework. We validate that the retrieval mechanism consistently selects relevant context and provide representative (query, retrieved tweets, response) examples in Appendix D.3. We investigate the sensitivity of our results to the choice of embedding model and retrieval depth in Appendix D. The temporal pattern of detected shifts is highly robust to both choices.

Our query set design constitutes a natural experiment. The pandemic arose from epidemiological processes entirely independent of Congresspersons' opinions, establishing temporal precedence and eliminating reverse causation. We selected public health queries due to their direct relevance to COVID-19, which has previously been tied to political opinion shifts (Kerr et al., 2021; Redbird et al., 2022). The other query sets serve as controls: candy & chocolate queries (or "orthogonal queries") capture generic temporal drift (shifting communication patterns, embedding artifacts, pipeline-specific factors, etc.), while general political queries additionally capture natural political evolution (opinion shifts, strategic repositioning, party messaging changes, etc.). If agent behavior with respect to public health queries exhibit temporal changes concentrated around COVID-19's onset that substantially exceed both baselines, this specificity supports a hypothesized causal interpretation.

**Temporal Gaussian blobs** To characterize statistical properties of our proposed tests, we consider simulated data inspired by our motivating setting. The data generation process creates paired observations across two timepoints for agents from two distinct classes (Figure 2). For a given agent, the process includes generating $R$ observations from each of $M$ noisy variations of an underlying latent structure.

In particular, each agent has a stable agent-specific effect

and a class-specific effect that combine to form its true latent perspective. Class 0 (the "signal class") exhibits temporal change: its class mean transitions from front-loaded (at $t = 1$) to back-loaded (at $t = 2$) across $p_s$ signal dimensions, with the shift magnitude controlled by effect size $\tau \in [0, 1]$. Class 1 (the "null class") has no temporal change. These latent perspectives are then randomly transformed in $p$ dimensions (differently for each query). Finally, observations are noisy realizations from a Gaussian centered around the randomly transformed latent perspectives.[3]

The simulation is governed by: (i) effect size $\tau \in [0, 1]$ controlling the magnitude of temporal shift, (ii) population size $N$ with balanced class assignment ($\pi = 0.5$), (iii) number of queries $M$, (iv) number of observations per query response $R$. This design enables evaluation of power and Type-I Error under varying signal-to-noise ratios and data complexities. Distributional details are provided in Appendix A.2.

### 3.2. Agent-level Perspective Shift

Let $\left\{ \left( f_n^{(t)} : n \in [N] \right) \right\}_{t \in [T]}$ denote $N$ agents over $T$ different timepoints. For a fixed $t, t'$ query set $Q$, and embedding function $g$, we test whether a particular agent $n$ changes between timepoints:

$$H_0 : f_n^{(t)} \overset{\mathcal{D}}{=} f_n^{(t')} \text{ against } H_A : f_n^{(t)} \overset{\mathcal{D}}{\neq} f_n^{(t')}. \quad (2)$$

Our proposed agent-level test (TDKPS) computes the Euclidean distance between TDKPS representations at timepoints $t$ and $t'$ as the test statistic, $||\psi_i^{(t)} - \psi_i^{(t')}||_F$. We assess significance via permutation: pooling the agent's response replicates from both timepoints, randomly redistributing them, and re-embedding the permuted data using

---

[3]Gaussians can approximate iterative (Wang et al., 2025) and recursive (Shumailov et al., 2024) functions of generative models.

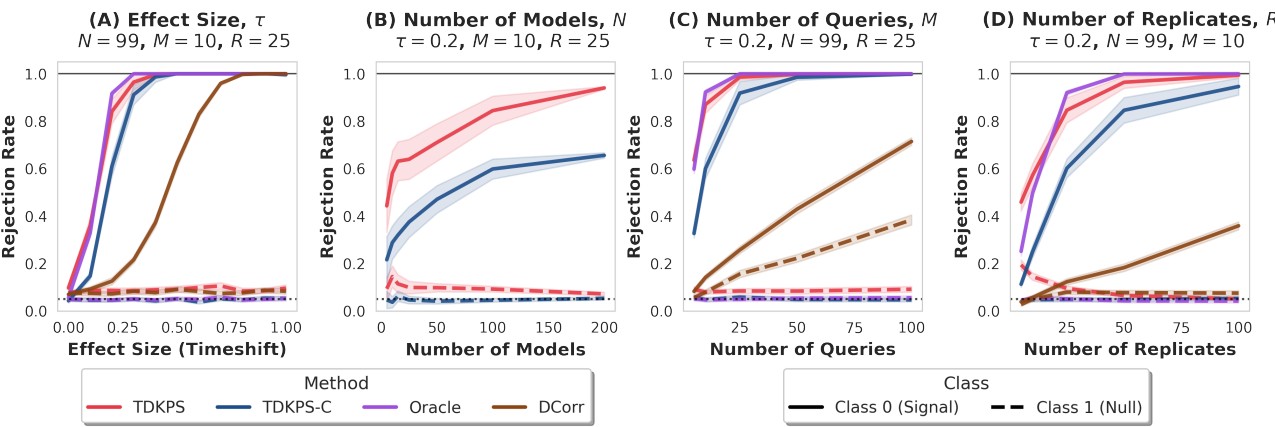

*Figure 3.* **Four simulations demonstrate strengths of `TDKPS`.** Power curves show mean rejection rates across 50 trials. Shaded regions: 95% confidence intervals; horizontal dotted line: nominal $\alpha = 0.05$. **(A)** Power increases with effect size. **(B)** More agents improve embedding stability (only `TDKPS` varies; other methods ignore pairwise structure). **(C)** Additional queries improve power. **(D)** Replicates substantially impact `TDKPS` power and validity.

the original fixed basis to isolate the effect of permuting the agent's response replicates from confounding changes in the embedding geometry. This generates a null distribution of within-agent distances to which we can compare the observed test statistic. Complete algorithmic details of our proposed test and its computational complexity are provided in Appendix B.1.

While our test is the first test designed explicitly to detect changes in the blackbox multi-agent system setting, we compare against two general baselines. First, an `oracle` that leverages unavailable knowledge (true orthogonal matrices, signal subspace dimensions, and generative agent structure) to untransform data, estimate and residualize agent-specific effects, then applies Hotelling's $T^2$ test (Hotelling, 1931). Second, a non-parametric `DCorr` baseline that tests independence between response distributions across timepoints for each response separately, then aggregates $p$-values using Fisher's method (Székely & Rizzo, 2004; Fisher, 1925). Complete algorithmic details are in Appendix B.

**Simulated Results**    We conducted systematic simulations varying four key parameters with Type I error tolerance $\alpha = 0.05$ (Figure 3). Each panel reports rejection rates for the signal class (solid lines, exhibiting true temporal differences) and null class (dashed lines, no differences). Parameter settings approximate or exceed the difficulty of our congressperson data (see Appendix A.3.1).

`TDKPS` consistently achieves near-optimal statistical power, closely approximating `oracle` and substantially outperforming `DCorr` across all conditions. While `TDKPS` exhibits slight Type I error inflation (5–10% for the null class), this remains stable and decreases with additional replicates (Figure 3D). In contrast, `DCorr`'s Type I error increases systematically with query count (Figure 3C), sug-

gesting specificity issues. Notably, even with just 5 agents, `TDKPS` power at effect size 0.2 nearly quadruples that of `DCorr` ($< 0.15$). Overall, `TDKPS` provides favorable Type I-Type II error tradeoffs for detecting agent-level behavioral changes. A sample-splitting variant of `TDKPS` (`TDKPS-C`; Appendix B.1.1) achieves nominal Type I error control at the cost of a modest reduction in power that diminishes with increasing $R$.

**Real Data Results**    Figure 4 studies our system of $N = 99$ agents across $T = 14$ timesteps between 2018 and 2024, for public health (Figure 4(A)), general political (Figure 4(B)), and null queries (Figure 4(C)). For visualization simplicity, we project the 3-dimensional[4] `TDKPS` embeddings via `LDA` along the discriminant direction which maximally separates Republicans (red) and Democrats (blue) (Figure 4I) (Fisher, 1936). Agent-level dynamics are indicated by faint lines; within-party averages are indicated in bold.

For each agent and at each $(t, t+1)$ pair, we use our agent-level test to determine if an agent's perspective has shifted. Faint lines indicate the empirical cumulative distribution function (CDF) of $p$-values for each timestep-pair across the agents (Figure 4II). We then group $(t, t+1)$ based on whether $t+1$ is within the first two years of COVID-19 (yellow; approximately April, $2020 -$ April, 2022) or not (black), and compute the empirical average (bold lines). Deviation from the CDF of a uniform distribution (the line $y = x$, dashed) indicates a perspective shift from $t$ to $t+1$ with respect to the query set. We note that the agents changed under all query sets and note our test does not achieve a nominal Type-1 Error rate in the simulation setting that matches our experiment ($R = 25$). We opted for an inflated

---

[4]`TDKPS` dimension determined via maximizing the profile likelihood of the eigenvalues from CMDS (Zhu & Ghodsi, 2006).

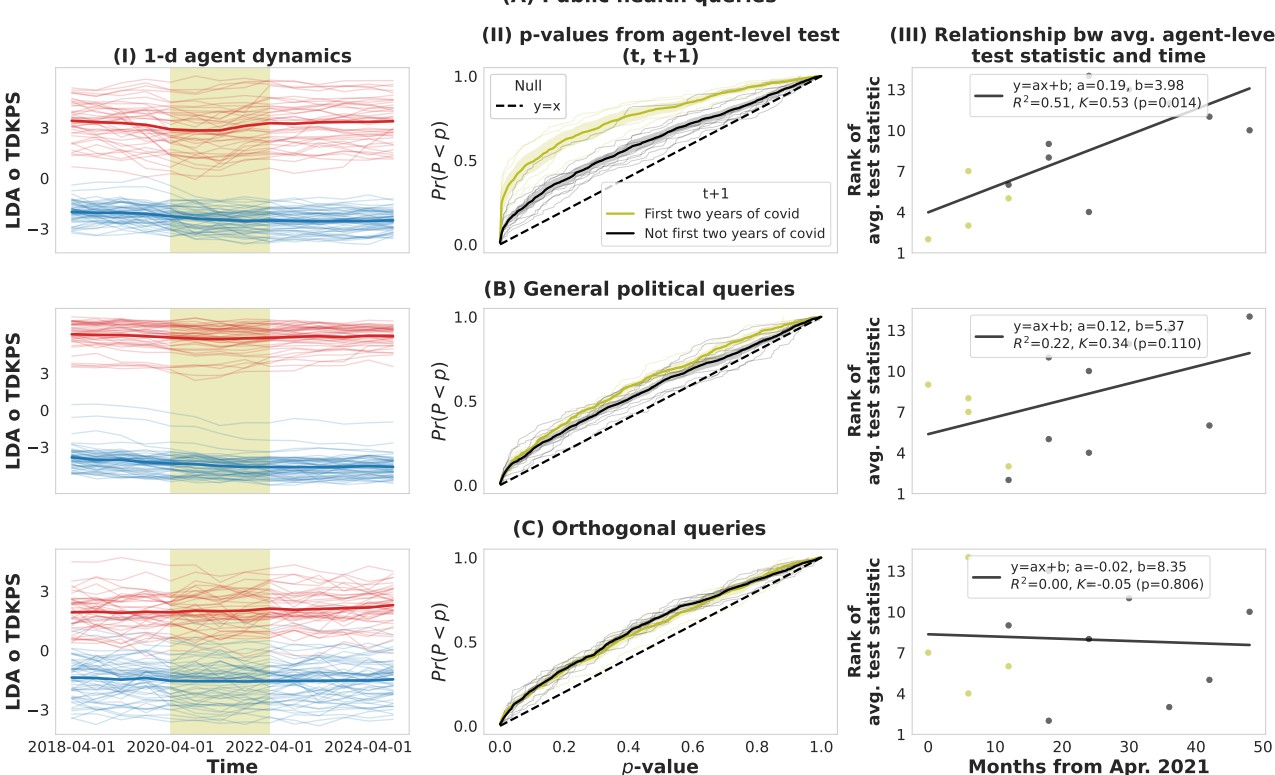

*Figure 4.* **Agent-level behavior shift after COVID-19 onset most pronounced for public health queries. (I)** TDKPS embeddings projected via LDA to separate Republicans (red) and Democrats (blue); faint lines show individual trajectories, bold lines show party averages. Yellow shading: first two years post-COVID-19 onset. **(II)** Empirical CDFs of $p$-values from agent-level tests, grouped by COVID-19 proximity (yellow/black). Deviation from diagonal indicates temporal shifts. **(III)** Rank of average TDKPS distance vs. temporal distance from April 2021. **(A)** Public health queries ($K = 0.51$, $p = 0.014$), **(B)** general political queries ($K = 0.34$, $p = 0.11$), **(C)** orthogonal queries ($K = 0.01$, $p = 0.956$).

Type-I error rate over a $4\times$ increase in computational costs.

We observe no particular temporal pattern between the first two years of COVID-19 and other years when investigating the multi-agent system with the null queries, and only slight differences for general political topics. For public health queries, however, we observe a significant trend: the four most extreme (i.e., the "sharpest") CDFs correspond to the bi-annual change points in the two year window following the onset of COVID-19.

To formalize this observation, we plot the rank of the average TDKPS embedding difference between consecutive timepoints (i.e., our agent-level test statistic, Figure 4C), where larger average differences correspond to lower ranks. The $x$-axis represents the temporal distance of $t + 1$ from the midpoint of our two-year post-COVID-19 onset window (approximately April 2021).

For public health queries, we observe strong correlation between rank and temporal proximity to this window (Kendall's tau $(K) = 0.51$, $p=0.014$): the largest behavioral shifts cluster tightly around COVID-19's onset. This tempo-

ral pattern is absent for orthogonal queries ($K = 0.01$, $p = 0.956$) and substantially weaker for general political queries ($K = 0.34$, $p = 0.11$). This differential specificity has a noteworthy implication: generic temporal drift would appear in candy queries, and natural political evolution would appear equally in general political queries, but we observe the effect concentrated in public health during the pandemic window.

**Computational complexity** While our agent-level test is performant, it is not without limitations for aggregate comparisons. Assuming the kernel matrix computation is amortized, we cannot avoid updating a distance matrix and then re-embedding it for each permutation. This results in a cost per-agent that scales with $\mathcal{O}(B \cdot (N \times T) \cdot M \cdot p + (N \times T)^2 \cdot d)$. With $B$, $M$, and $p$ typically very large, the first term is expensive to compute per-agent and can be prohibitively costly when computing it for all agents.

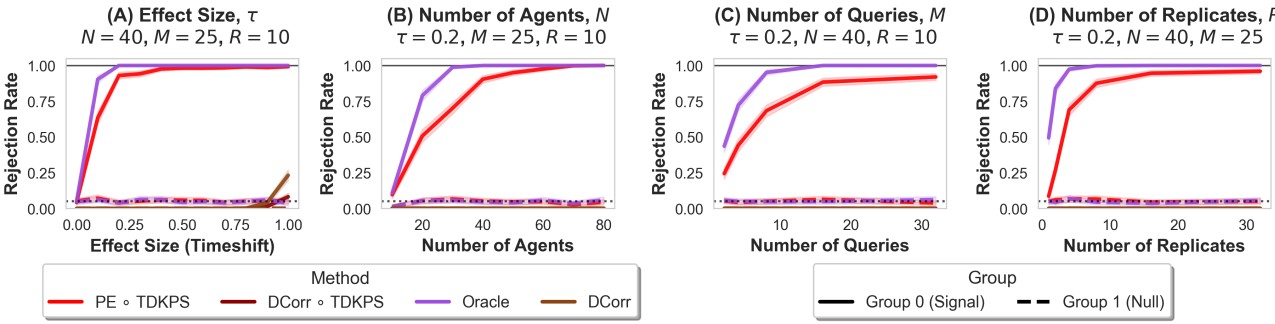

*Figure 5.* **Simulations demonstrate strengths of `PE ∘ TDKPS`.** Power curves show mean rejection rates across 200 trials (40 for **D**). Shaded regions: 95% confidence intervals; horizontal dotted line: $\alpha = 0.05$. Power increases with **(A)** effect size, **(B)** agents, **(C)** queries, and **(D)** replicates for `PE ∘ TDKPS`. Type I error of `PE ∘ TDKPS` remains $\approx \alpha$ across all conditions.

### 3.3. Group-level Perspective Shift

In multi-agent systems, some agents can be grouped together – by the set of tools an agent has access to, the regions of the environment they are interacting with, etc. Methods that incorporate this structure can provide more computationally efficient inference on aggregate agent behavior.

Let $\left\{ \left( f_n^{(t)} : n \in [N] \right) \right\}_{t \in [T]}$ denote $N$ agents over $T$ time-points, where each agent belongs to one of $L$ groups. For timepoints $t$ and $t'$, query set $Q$, and group $\ell \in [L]$, we test whether agents within group $\ell$ exhibit systematic temporal changes:

$$H_0 : F_\ell^{(t)} = F_\ell^{(t')} \text{ against } H_A : F_\ell^{(t)} \neq F_\ell^{(t')}, \quad (3)$$

where $F_\ell^{(t)}$ denotes the distribution of responses for agents in group $\ell$ at timepoint $t$.

Our proposed group-level test (`PE ∘ TDKPS`) computes an energy distance statistic between the empirical distributions of TDKPS embeddings at the two timepoints. We employ a paired permutation test: for each agent $n$ in group $\ell$, we randomly swap its timepoint indices with probability 0.5, recompute the test statistic using the same distance matrix, and repeat to generate a null distribution. This preserves marginal distributions and the paired nature of the data while breaking temporal pairing under the null. Our group-level test addresses the computational inefficiencies of the agent-level test; per-permutation, we perform only $\mathcal{O}(N^2)$ operations, which is cheaper than just a single agent-level test whenever $N < T \cdot M \cdot p$. This efficiency is achieved because the group-level test leverages across TDKPS embeddings, rather than re-embedding, for each permutation. Complete algorithmic details are provided in Appendix C.

We compare against three baselines. First, an `oracle` that untransforms data to the signal subspace, computes within-agent temporal differences, and applies paired Hotelling's $T^2$ test. Second, an unpaired `DCorr` baseline that pools agents from both timepoints and tests independence be-tween concatenated response vectors and timepoint labels. Third, `DCorr ∘ TDKPS` applies the same unpaired distance correlation test on TDKPS embeddings rather than raw responses. Complete algorithmic details and computational complexities are provided in Appendix C.

**Simulated Results** We conducted systematic simulations for `PE ∘ TDKPS` with challenging settings (SNR $\ll 0.02$; Figure 5, Appendix A.4). `PE ∘ TDKPS` consistently achieves near-optimal power, closely approximating `oracle` and dramatically outperforming both `DCorr` and `DCorr ∘ TDKPS` across all effect sizes (Figure 5A). In higher-difficulty regimes (Figures 5B–D), `DCorr` and `DCorr ∘ TDKPS` exhibit negligible power while `PE ∘ TDKPS` maintains strong performance. All methods maintained appropriate Type I error control ($\approx \alpha$) for the majority of settings.

**Real Data Results** We validate that our group-level test produces inference consistent with the more computationally intensive agent-level approach. Figure 6 presents three complementary analyses. Figure 6(A) shows the temporal dynamics of normalized test statistics for Republicans (R), Democrats (D), and all agents combined (ALL) across consecutive timepoint pairs. The group-level test (dashed lines) closely tracks the average agent-level test statistics (solid lines), with both approaches identifying elevated temporal change during the first two years of COVID-19 (shaded region) for public health queries.

Figure 6(B) quantifies the agreement between methods by comparing group-level $p$-values to combined agent-level $p$-values (via Fisher's method (Fisher, 1925)) across all timepoint pairs and groups. We observe strong correlations for Republicans (Kendall's tau ($K$) $= 0.69$, $p = 0.001$), Democrats ($K = 0.43$, $p = 0.043$), and all agents combined ($K = 0.62$, $p = 0.005$), demonstrating that the group-level test produces substantively similar inference to aggregated agent-level tests.

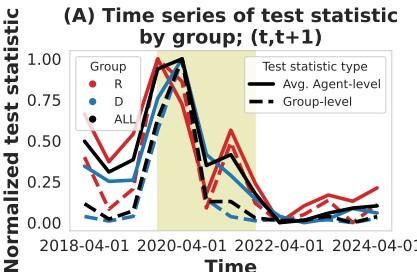

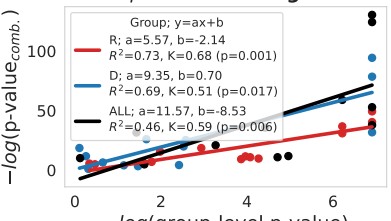

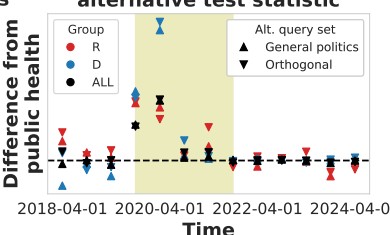

*Figure 6.* **Group-level test validation and agreement with agent-level inference.** **(A)** Temporal dynamics of normalized test statistics across consecutive timepoint pairs for Republicans (R), Democrats (D), and all agents (ALL). Both methods identify elevated temporal change during the COVID-19 period for public health queries. **(B)** Correlation between group-level $p$-values and combined agent-level $p$-values (Fisher's method) across timepoint pairs and groups, demonstrating strong agreement between methods. **(C)** Difference between public health group-level test statistics and alternative query sets (general politics, orthogonal) across time, reproducing the differential temporal pattern observed in agent-level analysis. A positive difference means public health queries induced a larger change.

Figure 6(C) reproduces the differential temporal pattern observed in our agent-level analysis: during the COVID-19 period (shaded), public health queries exhibit larger group-level test statistics compared to general political and orthogonal queries, while this difference disappears outside the COVID-19 window. This pattern again confirms that the group-level test captures the same topic-specific temporal dynamics identified by the agent-level approach, validating its use as a computationally efficient alternative for aggregate inference. Together, these results establish that both testing frameworks converge on the same substantive conclusions regarding temporal behavioral changes, with the group-level test providing orders of magnitude computational savings for aggregate agent-level questions.

## 4. Discussion

We introduced the Temporal Data Kernel Perspective Space (TDKPS) framework for detecting perspective shifts in multi-agent systems in the black-box setting. Our approach embeds agents into a low-dimensional Euclidean space that respects temporal dependencies and enables hypothesis testing at both agent and group levels. Through simulations, we demonstrated that TDKPS-based approaches achieve near-optimal statistical power for detecting temporal changes across diverse experimental conditions. The computational efficiency of our group-level test enables scalable aggregate inference for large multi-agent systems.

Our real data analysis validates that our proposed methodology, from agent construction through TDKPS embedding to hypothesis testing, captures genuine agent dynamics without any explicit modeling of agent interaction nor update mechanics. Applied to 99 digital congresspersons across 14 timepoints (2018–2024), we detected behavioral shifts consistent with an effect of COVID-19 on public health opinions. Importantly, our method detects *aggregate black-box behavioral shifts*: it identifies that the observable input-

output behavior of agents changed, without isolating which internal or external component – the base model, retrieval database, tool access, or environment – was responsible. In settings where some components are held fixed (e.g., a fixed base model with changing retrieval context, as in our congressperson system), the source of change can be narrowed by design. More generally, localizing the cause of a detected shift requires controlled experiments or additional domain knowledge beyond the scope of the TDKPS framework itself.

The changes we detect are temporally aligned with the pandemic's onset, concentrated specifically in public health queries rather than control topics, and scale with proximity to the outbreak period. This specificity arises from the natural experiment structure of our analysis: COVID-19 emerged from epidemiological processes independent of congressional opinion, establishing temporal precedence and ruling out reverse causation. The inclusion of orthogonal (candy & chocolate) and partially-related (general political) query sets provides built-in controls – generic temporal drift would appear in the orthogonal queries, and natural political evolution would appear equally in the political queries, yet the effect concentrates in public health during the pandemic window. While our analysis does not manipulate an exogenous event through direct experimentation, these features of the design – exogeneity, temporal alignment, topic specificity, and graded controls – collectively support a causal interpretation in the sense of Hill (1965), and are consistent with established evidence that political opinions shifted in response to the pandemic (Kerr et al., 2021; Redbird et al., 2022). Our real data analysis therefore supports the conclusion that TDKPS-like frameworks are capable of monitoring multi-agent systems in the black-box setting, and that careful query set design coupled with natural experiment structure can support identification of aggregate causal effects.

**Limitations** Type I error control for agent-level tests remains imperfect: while TDKPS maintains rejection rates of 5–10%, achieving nominal $\alpha = 0.05$ requires increasing response replicates. This reflects the challenge of constructing precise null distributions when permuting individual agents alters the kernel matrix structure, affecting all agents' embeddings through the shared representation. Type I error inflation may be able to be mitigated by using null distributions where the user knows no ("meaningful") change has taken place. We note that a sample-splitting variant of the agent-level test (Appendix B.1.1) achieves approximately nominal Type I error control by using separate subsets of replicates for basis estimation and test statistic computation, at the cost of a modest reduction in power. Other methods to control the size may better maintain power.

As highlighted by our inclusion of the null queries, query set dependence is fundamental – but poorly understood. In particular, we lack principled guidance for what constitutes a "sufficiently expansive" query set for a given application domain. Results are sensitive to query selection, particularly when queries probe narrow semantic regions or exhibit high inter-query correlation. We investigate the effect of signal sparsity in Appendix B.1.2, finding that the test's sensitivity is governed by total signal rather than the number of affected queries. Automated query set construction, for instance via LLM-based diversity-seeking generation or red-teaming, is a promising direction for addressing this dependence in practice.

Our framework treats query-response pairs as independent samples, appropriate for our congressperson application where agents respond to isolated queries. However, many modern agents exhibit context-dependent behaviors: maintaining conversation state, adapting responses based on dialogue history, or showing prompt-order sensitivity. For such systems, modeling temporal changes would require extensions that account for sequential dependencies.

**Future work** Two validation directions would strengthen our noteworthy conclusion. First, the validation of our method was limited to a particular class of simulation settings. Generalizing the simulations to improve our understanding of inference in TDKPS is necessary. Along these lines, theoretical analysis of the proposed tests may provide asymptotic and/or finite-sample guarantees for associated Type I and Type II error. Second, our real-data analysis examines one multi-agent system responding to one exogenous event. Applying TDKPS to additional diverse agent architectures, domains, and natural experiments would further confirm its generality.

Our framework extends naturally to richer temporal modeling. Continuous-time formulations could track agent trajectories through embedding space, enabling detection of gradual drift versus abrupt shifts, change point estimation, or forecasting. Multi-timepoint extensions could test for monotonic trends, periodic patterns, or complex temporal dynamics. Incorporating causal inference frameworks with controlled interventions could also strengthen attributions of behavioral change.

Our current hypothesis tests address what might be termed "zeroth-order" change detection: whether an agent's behavioral distribution has shifted at all. In settings where agents maintain internal state or memory, some behavioral change may be expected and even desirable, making a "first-order" formulation necessary, one that distinguishes normal state-driven drift from anomalous shifts, typically requiring a burn-in period to establish a baseline rate of change. Extending TDKPS to this regime is an important direction for monitoring stateful agents.

Establishing semantic interpretations of TDKPS dimensions would bridge geometric and semantic perspectives. By aligning embedding dimensions with interpretable axes through regression against external covariates, correlation with query features, or supervised alignment to human judgments, transforming TDKPS from purely relational geometry to semantically grounded representations. Such grounding would elevate the framework from detecting that an agent *has* drifted to characterizing *how*: distinguishing benign drift (paraphrasing, stylistic variation) from consequential drift along directions that correlate with degraded task performance, safety violations, or other failure modes. This could open the door to targeted, interpretable monitoring of multi-agent systems, surfacing the few directions that matter as a scalable safeguard for deployed agentic systems.

Our pipeline-level sensitivity analysis (Appendix D) provides initial evidence that TDKPS-based inference is robust to the choice of embedding model (Spearman $\rho > 0.99$ across four architectures) and moderately robust to retrieval depth ($\rho > 0.81$), and we confirm that the retrieval mechanism consistently selects relevant context across all query conditions (Appendix D.3), mitigating concerns that detected behavioral shifts could be driven by uninformative retrieved content. A more comprehensive investigation across additional pipeline configurations and application domains remains an important direction.

## Impact Statement

The framework developed herein, the Temporal Data Kernel Perspective Space (TDKPS), is the first to enable principled statistical inference on general agent dynamics in black-box multi-agent systems. We believe TDKPS and the broader topic of monitoring multi-agent dynamics will become increasingly important as agentic environments become more ubiquitous. Our methods enable detection of behavioral

changes in deployed AI systems, which is critical for safety monitoring, alignment verification, and understanding opinion dynamics in multi-agent environments. These capabilities could be applied beyond our intended use cases, including analysis of human-AI interaction patterns or political discourse dynamics. However, we developed this work to advance safe and interpretable multi-agent systems, and believe establishing principled frameworks for behavioral monitoring will be essential as agent deployment continues to scale.

## Acknowledgements

We would like to thank Tianyi Chen, Brandon Duderstadt, Teresa Huang, and Carey Priebe for their helpful comments throughout the development of this manuscript. Data wrangling and pre-processing, codebase optimization, and determining the query set for the congressperson agents were assisted with Claude 4.5 Sonnet (Anthropic, 2025) and GPT-5 (OpenAI, 2025). All model contributions were reviewed and validated by the authors, and all analytical / editorial decisions, interpretations, and conclusions are solely those of the authors.

We gratefully acknowledge funding from Defense Advanced Research Projects Agency (DARPA) Artificial Intelligence Quantified (AIQ) award number HR00112520026.

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

# A. Data

## A.1. Query generation

We generated 100 different queries for each of the topics {public health, general politics, candy & chocolate} with ChatGPT. The three prompts were:

1. **Public health.** We are doing a project in which we will be asking digital twins of politicians (i.e. agents) for their opinions on a variety of topics relating to public health that have been relevant between the years of 2016 and the present (2025). To do so, we would like a set of 100 queries which explore the public health landscape. Some of these queries might overlap, in that they might approach the same general topic area but from different angles. Each query should be a part of exactly one of the following subtopics: 1) Health Communication & Misinformation, 2) Science and Society, 3) Risk Perception & Public Engagement, and 4) Health Policy & Governance. The 100 queries should be evenly distributed across the four subtopics (i.e., 25 queries per subtopic). Please produce your response as a csv, with columns for query number, the associated subtopic, and the query itself.

2. **General politics.** We are doing a project in which we will be asking digital twins of politicians (i.e. agents) for their opinions on a variety of topics that have been relevant between the years of 2016 and the present (2025). To do so, we would like a set of 100 queries which explore the general political landscape. Each query should be a part of exactly one of the following subtopics: 1) Immigration, 2) The Economy, 3) LGBTQ Rights, and 4) Climate Change. The 100 queries should be evenly distributed across the four subtopics (i.e., 25 queries per subtopic). Please produce your response as a csv, with columns for query number, the associated subtopic, and the query itself.

3. **Candy & chocolate.** We are doing a project in which we will be asking digital twins of politicians (i.e. agents) for their opinions on a variety of topics relating to candy & chocolate that have been relevant between the years of 2016 and the present (2025). To do so, we would like a set of 100 queries which explore the candy & chocolate landscape. Each query should be a part of exactly one of the following subtopics: 1) Chocolate, 2) Frozen Treats, 3) Taffy & Toffee, and 4) Fruit-flavored Gummies. The 100 queries should be evenly distributed across the four subtopics (i.e., 25 queries per subtopic). Please produce your response as a csv, with columns for query number, the associated subtopic, and the query itself.

## A.2. Simulation settings

For agent $n$ at timepoint $t$, response $m$, and replicate $r$, the observed data are generated as:

$$x_{nmr}^{(t)} = \mathcal{O}_m \left( \mu_{y_n}^{(t)} + \eta_n \right) + \varepsilon_{nmr}^{(t)}$$

where:

- $\mathcal{O}_m \in \mathbb{R}^{p \times p}$ is a random response-specific orthogonal matrix,

- $\mu_{y_n}^{(t)}$ is the class $y_n$ mean at timepoint $t$,

- $\eta_n$ is a stable agent-level effect for agent $n$,

- $\varepsilon_{nmr}^{(t)} \overset{iid}{\sim} \mathcal{N}\left(0_p, \sigma_\epsilon^2 I_p\right)$ is independent measurement noise, ensuring that while responses are stochastically equivalent for a given agent, realizations will differ.

**Class assignment**     Agents are randomly assigned to one of two classes $y_n \overset{iid}{\sim} \text{Bern}(\pi = 0.5)$, where $y_n = 0$ is the "signal class" (the class with a differing distribution from $t = 1$ to $t = 2$) and $y_n = 1$ is the "null class") the class where agents do not have a differing distribution from $t = 1$ to $t = 2$).

**Generating latent perspectives**     For class 0 agents ($y_n = 0$), the population mean exhibits temporal dynamics controlled by an effect size parameter $\tau \in [0, 1]$, where for dimension $j \in [p]$ where $p_s$ is the number of signal dimensions, $\alpha = 1.0$ is

the scale parameter, and $\beta = -0.5$ is the decay rate:

$$\mu_0^{(t)}[j] = \begin{cases} \alpha \exp(-\beta j) & j \leq p_s, \text{ if } \tau = 0 \text{ or } t = 1 \\ \alpha \gamma(\tau)[(1-\tau)\exp(-\beta j) + \tau \exp(-\beta(p_s - 1 - j))] & j \leq p_s, \text{ if } \tau > 0 \text{ and } t = 2 \\ 0 & j > p_s \end{cases}$$

Here, $\tau$ is the effect size, and $\gamma(\tau)$ is a normalization constant to ensure that the signal magnitude across timepoints is constant. When $\tau = 0$, the signal profile for class 0 is frontloaded across both timepints (the null hypothesis is true, no signal change). When $\tau > 0$, the signal transitions from front-loaded at $t = 1$ to back-loaded at $t = 2$.

The null class has a fixed class-mean of 0; i.e., $\mu_1^{(t)} = 0_p$.

Each agent $n$ (regardless of class) has a persistent trait vector $\eta_n$ for all signal dimensions:

$$\eta_n[j] \overset{iid}{\sim} \begin{cases} \mathcal{N}(0, \sigma_{\mathcal{I}}^2) & j < p_s \\ 0 & j \geq p_s \end{cases}$$

These agent effects remain constant across timepoints, response modalities, and replicates, representing stable agent-specific characteristics.

**From perspectives to responses**    Response-specific orthogonal matrices $\mathcal{O}_m \overset{iid}{\sim} \text{Unif}\{SO(p)\}$ are sampled uniformly from the special orthogonal group $SO(p)$ for each response $m$. These matrices take an agent-specific latent perspective vector $\mu_{t,y_n} + \eta_n$ and distribute this latent perspective uniquely for responses $m$ across all feature dimensions.

These response vectors are then augmented by measurement-specific noise $\varepsilon_{nmk}^{(t)} \overset{iid}{\sim} \mathcal{N}(0_p, \sigma_\epsilon^2 I_p)$ for agent $n$ at time $t$ and response $m$ replicate $r$.

### A.3. Agent-level simulation parameters and hyperparameters

Unless otherwise specified, simulations used the following global settings: the number of features was $p = 200$, the scale $\alpha = 1.0$, the decay rate $\beta = 0.5$, the class assignment probability was $\pi = 0.5$, a decision threshold $\alpha = 0.05$, and permutation tests were conducted with $B = 1000$ permutations for TDKPS and DCorr. The number of components for TDKPS was selected automatically via elbow selection (Chung et al., 2019; Zhu & Ghodsi, 2006). Both the individual-level and measurement variance were $\sigma_{\mathcal{I}}^2 = \sigma_\epsilon^2 = \frac{1}{2}$. Each simulation is repeated for 50 independent trials.

**Effect Size**    We first examined whether methods could detect stronger individual differences by varying the timeshift (effect size) parameter, which controls the magnitude of temporal displacement in agent responses (Figure 3A). We varied the effect size parameter:

$$\tau \in \{0.0, 0.143, 0.286, 0.429, 0.571, 0.714, 0.857, 1.0\},$$

while holding other parameters fixed at: $n = 99$ agents, $p_s = 5$ signal dimensions, $M = 10$ queries, and $R = 25$ replicates per query. This setting was chosen to closely match our real-data digital congressperson investigation, in terms of the number of agents (we have 99 congresspersons) and computational feasibility (we investigated 25 response replicates per congressperson, per query).

As expected, all methods showed monotonically increasing power as effect sizes grew. TDKPS achieved near-optimal power, closely tracking the oracle baseline and substantially outperforming DCorr, particularly at moderate effect sizes (0.2 - 0.6). Both TDKPS and DCorr tend to show slight inflation of the type I error rate, with rejection rates hovering around 0.05 – 0.10.

**Number of agents**    Finally, we examined whether observing more agents in the broader population improves our ability to make inferences about any single agent (Figure 3B). A core advantage of TDKPS is that it first embeds agents by positioning them in a shared latent space; this intuitive, shared representation is the motivation of the TDKPS embedding. However, manipulating a agent across timesteps (e.g., permuting the query replicates across time as in TDKPS) alters the corresponding

rows/column of the distance matrix for that agent, and consequently, the stability of the subsequent embeddings. We varied the number of agents:

$$n \in \{2, 5, 10, 20, 50, 100\}$$

while holding other parameters fixed at: $p = 200$ total dimensions, $p_s = 5$ signal dimensions, $M = 10$ queries, $R = 25$ replicates per query, and $\tau = 0.2$ effect size. Class assignments $y_n \overset{iid}{\sim} \text{Bern}(0.5)$ ensure approximately balanced representation of signal and null classes at each sample size. This manipulation tests whether observing a larger population of agents improves the stability of individual-level inferences, particularly for methods like `TDKPS` that construct embeddings by positioning agents relative to the full ensemble.

This experiment is only investigated for `TDKPS`, since the other methods do not leverage a shared latent space. The results support this intuition. Increasing the number of agents improves `TDKPS` power (from 55% to 90%), demonstrating that population context is important for reliable agent-level inference with `TDKPS`.

**Number of queries**   A key practical question is whether observing agents across a more diverse set of queries improves our ability to detect agent differences (Figure 3C). We varied the number of queries:

$$M \in \{5, 10, 25, 50, 100\},$$

while holding other parameters fixed at: $n = 50$ agents, $p = 200$ total dimensions, $p_s = 5$ signal dimensions, $R = 25$ replicates per response, and $\tau = 0.2$ effect size. Each query $m$ has a unique random orthogonal matrix $\mathcal{O}_m \overset{iid}{\sim} \text{Unif}\{SO(p)\}$, ensuring that different queries probe different projections of the underlying latent perspective $\mu_{y_n}^{(t)} + \eta_n$.

Increasing the number of unique queries (i.e., with each query corresponding to a response from the agent) per agent dramatically improved power for all methods, with `TDKPS` and `oracle` reaching >90% power by 25 responses. `DCorr`'s Type I error rate increases rapidly as the number of queries increased, suggesting the method may overfit to spurious patterns arising at the query-level when given more data, likely due to the $p$-value aggregation procedure leveraged across queries. In contrast, `TDKPS` maintained stable Type I error rates, indicating more robust validity scaling to broader query contexts.

**Number of replicates**   Beyond diversity of queries, we also examined whether observing multiple response replicates to the same query aids detection (Figure 3D). We varied the number of replicates per query:

$$R \in \{5, 10, 25, 50, 100\}$$

while holding other parameters fixed at: $n = 50$ agents, $p = 200$ total dimensions, $p_s = 5$ signal dimensions, $M = 10$ queries, and $\tau = 0.2$ effect size. For each replicate $r$ of query $m$, independent measurement noise $\varepsilon_{nmr}^{(t)} \overset{iid}{\sim} \mathcal{N}\left(0_p, \frac{1}{2} I_p\right)$ is added to the rotated signal $\mathcal{O}_m(\mu_{y_n}^{(t)} + \eta_n)$, ensuring stochastic variation across replicates while maintaining the same underlying response structure.

Increasing replicates yields substantial power gains for `TDKPS` and `oracle`, approaching perfect detection by 60 replicates. This improvement likely reflects reduced measurement noise: averaging multiple replicate responses per query provides a more stable estimate of each agent's characteristic response pattern. `DCorr` shows modest gains to increasing replicates, which is surprising because these replicates are directly included for each of the $M$ query-level tests. The specificity of `TDKPS` improves rapidly with more response replicates to each query, likely due to the within-response permutation procedure across replicates providing more precise approximations of the null distribution.

Signal dimensionality We manipulated the dimensionality of the signal subspace (Figure 3D), which relates to signal-to-noise ratio in a nonlinear fashion: more signal dimensions increase SNR, but with diminishing returns (due to the exponential decay of the added signal). We varied the number of signal dimensions:

$$p_s \in \{3, 5, 8, 12, 20\}$$

while holding other parameters fixed at: $n = 50$ agents, $p = 200$ total dimensions, $M = 10$ queries, $R = 25$ replicates per query, and $\tau = 0.2$ effect size. The signal magnitude in dimension $j \le p_s$ follows an exponential decay $\alpha \exp(-\beta j)$ with $\alpha = 1.0$ and $\beta = 0.5$. The stable agent-level effects $\eta_n[j] \overset{iid}{\sim} \mathcal{N}\left(0, \sigma_{\mathcal{I}}^2\right)$ with $\sigma_{\mathcal{I}} = \sqrt{0.5}$ are non-zero only for $j < p_s$, while

$\eta_n[j] = 0$ for $j \geq p_s$. As $p_s$ increases, total signal-to-noise ratio increases with diminishing returns due to the exponential decay.

`TDKPS` achieves 60–80% power and remains relatively stable across signal dimensionalities, suggesting the method adapts reasonably well to the varying regimes. Interestingly, `TDKPS` power showed a slight decrease at higher dimensionalities (15–20 dimensions), potentially reflecting the dual effect of increased signal strength being offset by the increase in the subsequent embedding's complexity. `DCorr` showed very low power (less than 20%) across all conditions, indicating fundamental insensitivity to the types of structured individual differences present in these simulations.

### A.3.1. SIGNAL-TO-NOISE RATIO

**Agent-level tests**    The simulations were designed to be extremely challenging, with signal-to-noise ratios $\ll 0.05$ for all investigations except signal dimensionality. This is because otherwise, `TDKPS` showed near-perfect power for even minute effect sizes, which reduced the practical value of empirical simulation.

To see this, consider a loose upper bound on the SNR. The signal component for detecting temporal change is the shift in class mean from $t = 1$ to $t = 2$: $|\mu_0^{(2)}[j] - \mu_0^{(1)}[j]|$. The noise components include both measurement noise (variance $\sigma_\epsilon^2 = 0.5$ per dimension) and the stable agent effects $\eta_n[j]$ (variance $\sigma_\mathcal{I}^2 = 0.5$ per dimension), which are both mean 0 and thus only obscure the temporal signal.

For a crude upper bound on SNR, consider the most favorable case where signal accumulates coherently across $p_s = 5$ dimensions with maximum effect size (timeshift) $\tau = 1$ at a signal scale of 1.0. The maximum possible signal disparity between any signal dimensions from $t = 1$ to $t = 2$ is strictly $< 1$ (dimension 0), and exponentially decays for successive dimensions at rate $\beta$ (typically taken to be 0.5). The total signal power is therefore at most:

$$\text{Signal}(\tau) \leq \text{Signal}(\tau = 1) \ll p_s \times 1 = 5$$

The total noise variance across $p = 200$ dimensions (incorporating only measurement noise of variance $= 0.5$, and ignoring the agent-level variability with variance $= 0.5$) is $200 \times 0.5 = 100$. Thus:

$$\text{SNR}(\tau) \ll \frac{5}{100} = 0.05$$

This extremely crude upper bound becomes even more pessimistic when properly accounting for the exponential decay across signal dimensions, that this bound was constructed assuming the worst-possible effect size (timeshift) of 1.0 and is even lower for other effect sizes (most of our simulations use effect size of 0.2), and agent-specific variability.

### A.3.2. SPARSE SIGNAL STRUCTURE

To evaluate sensitivity to signal sparsity, we modify the simulation so that only a fraction $s \in (0, 1]$ of queries carry temporal signal. Specifically, for each of the $M$ queries, we independently assign it as a "signal query" with probability $s$. For signal queries, the class-specific temporal shift operates as in Section A.2; for non-signal queries, the class means are identical across timepoints (i.e., no temporal change is introduced for that query). We consider two regimes:

**Scaling signal.**    The per-query shift magnitude is held fixed at $\delta = 1.0$, so the total signal energy grows linearly with $s$. This regime tests whether the method can detect weak, distributed signal when few queries are affected.

**Constant signal.**    The total signal energy is held fixed by setting the per-query shift magnitude to $\delta/\sqrt{s}$, where $\delta = 1.0$ is the baseline shift at $s = 1.0$. This regime tests whether the method is sensitive to how signal is distributed across queries when the aggregate signal strength is constant.

In both regimes, all other parameters are held fixed at: $N = 25$ agents, $M = 20$ queries, $R = 25$ replicates, $p = 50$ total dimensions, $p_s = 5$ signal dimensions. Results are estimated from 50 Monte Carlo replicates with $B = 200$ permutations per test and $\alpha = 0.05$.

### A.4. Group-level simulation parameters and hyperparameters

Unless otherwise specified, simulations used the following global settings: the number of features was $p = 200$, the scale $\alpha = 0.75$, the decay rate $\beta = 0.5$, the class assignment probability was $\pi = 0.5$, a decision threshold $\alpha = 0.05$, and

permutation tests were conducted with $B = 1000$ permutations for all methods. The number of components for methods using TDKPS embeddings was selected automatically via elbow selection (Chung et al., 2019; Zhu & Ghodsi, 2006). Both the individual-level and measurement variance were $\sigma_{\mathcal{I}}^2 = \sigma_{\epsilon}^2 = 1$. Each simulation is repeated for 500 independent trials.

**Effect Size**  We first examined whether methods could detect group-level temporal shifts by varying the timeshift (effect size) parameter (Figure 5A). We varied the effect size parameter:

$$\tau \in \{0.0, 0.1, 0.2, \ldots, 1.0\},$$

while holding other parameters fixed at: $N = 40$ agents per group, $p_s = 5$ signal dimensions, $M = 25$ queries, and $R = 10$ replicates per query.

As expected, all methods showed monotonically increasing power as effect sizes grew. `PE ∘ TDKPS` (Paired Energy on TDKPS embeddings) achieved near-optimal power, closely tracking the `oracle` baseline and substantially outperforming both `DCorr ∘ TDKPS` and standalone `DCorr`, which only attain any power at the largest effect sizes. All methods maintained appropriate Type I error control for the null group, with rejection rates near the nominal $\alpha = 0.05$ level.

**Number of queries**  We investigated whether observing groups across more diverse query contexts improves detection of group-level temporal patterns (Figure 5B). We varied the number of queries:

$$M \in \{2, 4, 8, 16, 32\},$$

while holding other parameters fixed at: $N = 40$ agents per group, $p = 200$ total dimensions, $p_s = 5$ signal dimensions, $R = 10$ replicates per query, and $\tau = 0.2$ effect size. Each query $m$ has a unique random orthogonal matrix $\mathcal{O}_m \overset{\text{iid}}{\sim}$ Unif$\{SO(p)\}$.

Increasing the number of queries yielded dramatic power improvements for all methods, with `PE ∘ TDKPS` and `oracle` achieving near-perfect detection by 16–32 queries. All methods maintained stable Type I error rates as query counts increased, indicating robust validity when scaling to broader query contexts. Both `DCorr` and `DCorr ∘ TDKPS` are uninformative for signal detection.

**Number of replicates**  We examined whether multiple response replicates to the same query improve group-level detection (Figure 5C). We varied the number of replicates per query:

$$R \in \{1, 2, 4, 8, 16, 32\}$$

while holding other parameters fixed at: $N = 40$ agents per group, $p = 200$ total dimensions, $p_s = 5$ signal dimensions, $M = 25$ queries, and $\tau = 0.2$ effect size. For each replicate $r$ of query $m$, independent measurement noise $\varepsilon_{nmr}^{(t)} \overset{\text{iid}}{\sim} \mathcal{N}(0_p, I_p)$ is added to the rotated signal.

Increasing replicates yielded substantial power gains for all methods, with `PE ∘ TDKPS` and `oracle` approaching perfect detection by 8–16 replicates. This improvement reflects reduced measurement noise: averaging multiple replicate responses per query provides more stable estimates of each agent's characteristic response pattern, which in turn improves the stability of the group-level comparisons. Type I error rates remained well-controlled across all replicate counts for all methods. Both `DCorr` and `DCorr ∘ TDKPS` are uninformative for signal detection.

**Number of agents**  Finally, we examined whether observing larger groups improves our ability to detect group-level temporal patterns (Figure 5D). We varied the number of agents per group:

$$N \in \{10, 20, 30, 40, 50, 60, 70, 80\}$$

while holding other parameters fixed at: $p = 200$ total dimensions, $p_s = 5$ signal dimensions, $M = 25$ queries, $R = 10$ replicates per query, and $\tau = 0.2$ effect size. Class assignments $y_n \overset{\text{iid}}{\sim}$ Bern$(0.5)$ ensure approximately balanced group sizes.

Increasing the number of agents per group dramatically improved power for all methods, with `PE ∘ TDKPS` rising from ~50% at 10 agents to >95% at 80 agents. The `oracle` showed even faster convergence to perfect power. This improvement reflects both the increased statistical power from larger sample sizes and the improved stability of the TDKPS embeddings when constructed from larger populations. Notably, all methods maintained proper Type I error control across all group sizes, with the null group showing rejection rates consistently near $\alpha = 0.05$. Both `DCorr` and `DCorr ∘ TDKPS` are uninformative for signal detection.

A.4.1. SIGNAL-TO-NOISE RATIO

**Group-level tests**   For the group-level tests, $\sigma_\epsilon^2 = \sigma_\mathcal{I}^2 = 1$, and the maximum possible signal at $\tau = 1$ at a signal scale of $0.75$ is $0.75$. By a similar argument to the above:

$$\text{Signal}(\tau) \leq \text{Signal}(\tau = 1) \ll p_s \times 1 = \frac{15}{4}$$

The total noise variance across $p = 200$ dimensions (incorporating only the measurement noise of variance $= 1$, and ignoring agent-level variability with variance $= 1$) is $200$; thus:

$$\text{SNR}(\tau) \ll \frac{}{200} < 0.02,$$

Which again is far overly conservative when properly accounting for the exponential decay across signal dimensions, that this bound was constructed assuming the worst-possible effect size (timeshift) of $1.0$ and is lower for other effect sizes (most simulations use effect size of $0.1$), and agent-specific variability.

# B. Agent-level tests

For a time-varying agent $f_n^{(t)}$ and $f_n^{(t')}$, we have the null hypothesis:

$$H_0 : f_n^{(t)} = f_n^{(t')} \text{ against } H_A : f_n^{(t)} \neq f_n^{(t')}. \tag{4}$$

For each agent, we observe independent response samples $X_{nmr}^{(t)} = g(f_n^{(t)}(q_m))_r$, whose distribution depends on the agent $n$, the query $m$, and the timestep $t$.

## B.1. TDKPS

The `TDKPS` agent-level test leverages the temporal data kernel perspective space embeddings described in Section 2 to construct a fully non-parametric geometric test of temporal change. This approach combines dimensionality reduction via classical multidimensional scaling with a paired permutation framework, operating in an interpretable low-dimensional embedding space rather than the high-dimensional ambient response space.

Let $\left\{ \left( \psi_n^{(t)} : n \in [N] \right) \right\}_{t \in [T]}$ be the TDKPS representations of $N$ agents across $T$ timepoints. Our goal is to detect if the TDKPS representation of agent $n$ at time $t$ is different from the TDKPS representation of agent $n$ at time $t'$. We pose this as a statistical hypothesis test:

$$H_0 : \psi_n^{(t)} \overset{\mathcal{D}}{=} \psi_n^{(t')} \text{ against } H_A : \psi_n^{(t)} \overset{\mathcal{D}}{\neq} \psi_n^{(t')}.$$

We emphasize that the latent TDKPS representations for agent $n$ at timepoints $t$ and $t'$ are dependent on the query set $Q$, the agent $n$ at other timepoints $t'' \notin \{t, t'\}$, as well as the agents for other agents $n' \neq n$ across all timepoints $t'' \in [T]$.

For agent $n$, a natural test statistic for this hypothesis is the Euclidean distance between TDKPS representations at timepoints $t$ and $t'$:

$$\delta_n = \left\| \psi_n^{(t)} - \psi_n^{(t')} \right\|_2,$$

where $\psi_n^{(t)}$ and $\psi_n^{(t')}$ are the TDKPS embeddings of agent $n$ at timepoints $t$ and $t'$ respectively.

To assess statistical significance under the null hypothesis of no temporal change, we employ a paired permutation test:

**Step 1: Compute observed statistic**   Fit the TDKPS estimator to obtain embeddings $\left\{ \psi_n^{(t)} \right\}$ for all agents and timepoints, and compute the observed distance $\delta_n$ for agent $n$.

**Step 2: Generate null distribution**   For permutation $b = 1, \ldots, B$:

(a) Pool agent $n$'s response replicates from both timepoints: $\{X_{nmr}^{(t)}, X_{nmr}^{(t')} : m \in \{1, \ldots, M\}, r \in \{1, \ldots, R\}\}$,

(b) Randomly redistribute replicates across the two timepoints, maintaining the same number of replicates per timepoint and response,

(c) Re-embed the data leveraging the permuted data for agent $n$ at timepoints $t$ and $t'$ and the original data for other agent/timepoint combinations, using the original fixed basis (projection matrix $V\Sigma^{-\frac{1}{2}}$ from the original SVD) to obtain the modified embeddings $\tilde{\psi}_n^{(t)}$ and $\tilde{\psi}_n^{(t')}$ for agent $n$ after paired permutation,

(d) Compute the distance between the modified embeddings after paired permutation: $\delta_n^{(b)} = \left\| \tilde{\psi}_n^{(t)} - \tilde{\psi}_n^{(t')} \right\|_2$.

**Step 3: Compute $p$-value** The p-value is computed as:

$$p = \frac{1 + \sum_{b=1}^{B} \mathbf{1}_{\left\{ \delta_n^{(b)} \geq \delta_n \right\}}}{1 + B}.$$

**Key properties** The `TDKPS` approach:

- Operates in an interpretable low-dimensional embedding space, facilitating geometric interpretation and visualization of temporal changes,

- Makes no parametric assumptions about the data distribution,

- Does not require knowledge of orthogonal matrices or signal dimensions,

- Uses a fixed basis for permuted embeddings to avoid three issues: (1) eliminates orthogonal non-identifiability by ensuring permuted embeddings remain in the same coordinate system as the original, and (2) prevents spurious changes in the embedding space that could arise from the permuted agent's altered role in the distance matrix structure (if an agent being permuted is particularly "influential", permuting its samples could yield a vastly different embedding; for instance, after permutation, fewer embedding dimensions are estimated), and

- Accounts for within-agent variability through the paired permutation mechanism.

**Computational complexity** The test begins with a one-time computation of TDKPS embeddings: constructing the distance matrix requires $\mathcal{O}((N \times T) \cdot M \cdot p \cdot R + (N \times T)^2 \cdot M \cdot p) = \mathcal{O}((N \times T)^2 \cdot M \cdot p)$ operations when $R < N \times T$, and computing the SVD to obtain $d$ components requires $\mathcal{O}((N \times T)^2 \cdot d)$ operations using truncated SVD (which is negligible when $d \ll p$). For each of $B$ permutations, the test then performs two primary operations: updating the distance matrix and re-embedding via the fixed basis. The distance update recomputes distances for the shuffled agent at the two timepoints to all other agent-timepoint combinations, requiring $\mathcal{O}((N \times T) \cdot M \cdot p)$ operations where $M$ is the number of queries and $p$ is the ambient response dimension. Re-embedding projects the updated distance matrix through the fixed basis ($V\Sigma^{-1/2}$ from the original SVD), requiring $\mathcal{O}((N \times T)^2 \cdot d)$ operations where $d$ is the TDKPS embedding dimension. The dominant per-permutation term depends on the regime: distance updates dominate when $M \cdot p > (N \times T) \cdot d$, while re-embedding dominates in the opposite regime. Overall per-agent complexity is $\mathcal{O}((N \times T)^2 \cdot M \cdot p + B \cdot ((N \times T) \cdot M \cdot p + (N \times T)^2 \cdot d))$ for $B$ permutations.

When testing all $N$ agents independently, the one-time embedding cost is amortized, but each agent requires $B$ distance matrix updates and re-embeddings. Total complexity becomes $\mathcal{O}((N \times T)^2 \cdot M \cdot p + N \cdot B \cdot ((N \times T) \cdot M \cdot p + (N \times T)^2 \cdot d))$. For typical scenarios where $B$ is large (e.g., $B = 1000$), the permutation costs dominate, yielding $\mathcal{O}(B \cdot N^2 \cdot T \cdot M \cdot p)$ when distance matrix updates are the bottleneck, or $\mathcal{O}(B \cdot N \cdot (N \times T)^2 \cdot d) = \mathcal{O}(B \cdot N^3 \cdot T^2 \cdot d)$ when re-embedding dominates. This cubic dependence on $N$ (resulting from applying $N$-independent agent-level tests) motivates more efficient aggregate-level approaches.

B.1.1. CALIBRATED AGENT-LEVEL TEST

The agent-level test described in Section B.1 exhibits mild Type I error inflation (5–10%) because permuting a single agent's replicates alters the kernel matrix, which in turn affects the embedding geometry used to compute the test statistic. This coupling between the permuted data and the embedding basis is the source of the inflation.

We propose a *sample-splitting* variant that breaks this dependence. For each agent $n$ at each timepoint $t$ and query $q_m$, we randomly partition the $R$ response replicates into two disjoint sets of size $\lfloor R/2 \rfloor$: a *basis set* $\mathcal{B}$ and a *test set* $\mathcal{T}$. The procedure is as follows:

1. **Estimate the embedding basis.** Using only the basis-set replicates $\mathcal{B}$, compute the averaged response matrices $\bar{X}_{n,\mathcal{B}}^{(t)}$, the block distance matrix $\tilde{D}_{\mathcal{B}}$, and the SVD-derived projection $V_{\mathcal{B}}\Sigma_{\mathcal{B}}^{-1/2}$.

2. **Compute the observed test statistic.** Using only the test-set replicates $\mathcal{T}$, compute the averaged response matrices $\bar{X}_{n,\mathcal{T}}^{(t)}$, the corresponding block distance matrix $\tilde{D}_{\mathcal{T}}$, and project via the *fixed* basis from Step 1 to obtain embeddings $\hat{\psi}_n^{(t)}$. The observed statistic is $\delta_n = \|\hat{\psi}_n^{(t)} - \hat{\psi}_n^{(t')}\|_2$.

3. **Generate the null distribution.** For each permutation $b = 1, \ldots, B$: pool agent $n$'s test-set replicates from both timepoints, randomly redistribute them, recompute $\tilde{D}_{\mathcal{T}}^{(b)}$, project through the same fixed basis, and compute $\delta_n^{(b)}$.

4. **Compute the $p$-value** as $p = (1 + \sum_{b=1}^{B} \mathbf{1}\{\delta_n^{(b)} \geq \delta_n\})/(1 + B)$.

Because the embedding basis is estimated from data that is never permuted, the projection geometry remains invariant across the null distribution, eliminating the coupling that causes Type I error inflation in the uncalibrated test.

The cost of this improved calibration is a reduction in the effective sample size for both the basis estimation and the test statistic, each of which uses $\lfloor R/2 \rfloor$ rather than $R$ replicates. This manifests as a modest decrease in statistical power, particularly at small $R$.

**Empirical comparison.** Table 1 reports rejection probabilities for the calibrated and uncalibrated tests across a range of replicate counts ($R \in \{10, 20, 50\}$), with $N = 25$, $M = 10$, $p = 50$, $p_s = 5$, $\alpha = 0.05$. Values are estimated from 100 Monte Carlo replicates with $B = 200$ permutations per test; $\pm$ denotes standard error.

*Table 1.* Rejection probabilities for the calibrated and uncalibrated agent-level tests. **Top**: $\tau = 0$ (Type I error for both classes). **Bottom**: $\tau = 0.5$ (power for Class 0; Type I error for Class 1).

| | Class 0 | | Class 1 | |
| --- | --- | --- | --- | --- |
| $R$ | Uncalibrated | Calibrated | Uncalibrated | Calibrated |
| | | $\tau = 0$ (Type I Error) | | |
| 10 | $0.072 \pm 0.008$ | $0.040 \pm 0.006$ | $0.098 \pm 0.009$ | $0.053 \pm 0.006$ |
| 20 | $0.069 \pm 0.008$ | $0.047 \pm 0.006$ | $0.070 \pm 0.007$ | $0.048 \pm 0.006$ |
| 50 | $0.070 \pm 0.007$ | $0.048 \pm 0.007$ | $0.072 \pm 0.008$ | $0.056 \pm 0.006$ |
| | | $\tau = 0.5$ (Power / Type I Error) | | |
| 10 | $0.828 \pm 0.026$ | $0.683 \pm 0.033$ | $0.102 \pm 0.009$ | $0.039 \pm 0.005$ |
| 20 | $0.905 \pm 0.017$ | $0.780 \pm 0.029$ | $0.073 \pm 0.007$ | $0.047 \pm 0.006$ |
| 50 | $0.971 \pm 0.009$ | $0.936 \pm 0.014$ | $0.071 \pm 0.008$ | $0.052 \pm 0.006$ |

Under the null ($\tau = 0$), the calibrated test achieves rejection rates near the nominal $\alpha = 0.05$ across all replicate counts, whereas the uncalibrated test consistently exceeds $\alpha$. Under the alternative ($\tau = 0.5$), the calibrated test retains high power that converges toward the uncalibrated test as $R$ increases: at $R = 50$, the power gap narrows to approximately 3.5 percentage points. The calibrated test thus offers a principled trade-off: nominal Type I error control at the cost of a modest and diminishing power reduction.

### B.1.2. SENSITIVITY TO SIGNAL SPARSITY

We investigate how the agent-level test performs when temporal signal is concentrated on a subset of queries rather than distributed uniformly. This is motivated by realistic scenarios in which an agent's behavior changes on only a small fraction of queries (e.g., an LLM-based agent that shifts its responses on a narrow topic while remaining stable elsewhere). We report results for both the uncalibrated and calibrated (sample-split; Appendix B.1.1) tests. Simulation parameters are detailed in Appendix A.3.2.

**Scaling signal.** Table 2 reports rejection probabilities when the per-query shift is fixed at $\delta = 1.0$ and total signal grows with the fraction of affected queries $s$.

*Table 2.* Rejection probabilities under **scaling signal**: per-query shift fixed at $\delta = 1.0$; total signal grows with $s$. Class 0 columns report power; Class 1 columns report Type I error. $\pm$ denotes standard error over 50 trials.

| $s$ | $\delta$ | Class 0 (Power) | | Class 1 (Type I Error) | |
|---|---|---|---|---|---|
| | | Uncalib. | Calib. | Uncalib. | Calib. |
| 0.05 | 1.00 | $0.083 \pm 0.013$ | $0.057 \pm 0.011$ | $0.073 \pm 0.010$ | $0.039 \pm 0.009$ |
| 0.10 | 1.00 | $0.116 \pm 0.018$ | $0.084 \pm 0.015$ | $0.074 \pm 0.013$ | $0.048 \pm 0.011$ |
| 0.25 | 1.00 | $0.192 \pm 0.028$ | $0.123 \pm 0.016$ | $0.054 \pm 0.011$ | $0.055 \pm 0.008$ |
| 0.50 | 1.00 | $0.584 \pm 0.039$ | $0.390 \pm 0.036$ | $0.070 \pm 0.013$ | $0.061 \pm 0.010$ |
| 1.00 | 1.00 | $0.928 \pm 0.019$ | $0.788 \pm 0.033$ | $0.050 \pm 0.011$ | $0.029 \pm 0.007$ |

Power increases monotonically with $s$: when only 5% of queries carry signal ($s = 0.05$, i.e., 1 of 20 queries), the total signal is weak and the calibrated test does not exceed the nominal level. This is expected –the test aggregates information across all queries, and with fixed per-query shift, extremely sparse signal provides insufficient aggregate evidence.

**Constant signal.** Table 3 reports rejection probabilities when total signal is held fixed by scaling the per-query shift as $\delta/\sqrt{s}$.

*Table 3.* Rejection probabilities under **constant signal**: total signal fixed; per-query shift scales as $\delta/\sqrt{s}$ with $\delta = 1.0$. Class 0 columns report power; Class 1 columns report Type I error. $\pm$ denotes standard error over 50 trials.

| $s$ | $\delta$ | Class 0 (Power) | | Class 1 (Type I Error) | |
|---|---|---|---|---|---|
| | | Uncalib. | Calib. | Uncalib. | Calib. |
| 0.05 | 4.47 | $0.924 \pm 0.020$ | $0.798 \pm 0.031$ | $0.071 \pm 0.009$ | $0.036 \pm 0.007$ |
| 0.10 | 3.16 | $0.918 \pm 0.018$ | $0.779 \pm 0.034$ | $0.071 \pm 0.011$ | $0.053 \pm 0.012$ |
| 0.25 | 2.00 | $0.871 \pm 0.025$ | $0.696 \pm 0.040$ | $0.042 \pm 0.010$ | $0.047 \pm 0.008$ |
| 0.50 | 1.41 | $0.915 \pm 0.023$ | $0.802 \pm 0.034$ | $0.067 \pm 0.012$ | $0.054 \pm 0.010$ |
| 1.00 | 1.00 | $0.923 \pm 0.021$ | $0.781 \pm 0.032$ | $0.056 \pm 0.011$ | $0.032 \pm 0.008$ |

When total signal is held constant, power is approximately flat across all sparsity levels: 0.70–0.80 for the calibrated test and 0.87–0.92 for the uncalibrated test. The agent-level test detects the temporal change equally well whether the signal is concentrated on a single query ($s = 0.05$) or distributed across all queries ($s = 1.0$). This demonstrates that the sensitivity of the TDKPS agent-level test is governed by *total signal* rather than the number of affected queries, an important practical consideration for query set design: a small, well-targeted query set can be as effective as a large one, provided the aggregate signal is sufficient.

## B.2. Oracle

The `oracle` represents an upper bound on performance by leveraging knowledge unavailable in practice. This oracle knows:

1. the true orthogonal matrices $\mathcal{O}_m$ for each response $m$,

2. that signal is concentrated in the first $p_s$ dimensions after untransforming, and

3. the generative agent structure (population means plus agent-specific random effects plus noise).

This oracle constructs a near-optimal parametric test by unrotating to the signal subspace and using the known agent structure to residualize agent-specific effects. While stronger oracles are possible (e.g., one with access to the true agent-specific effects $\alpha_n$, or oracles that pool data across agents), this represents the optimal strategy achievable using a agent's observed data.

Under the generative agent, testing Equation (4) is equivalent to testing whether the population means in the signal subspace differ between timepoints:

$$H_0 : \mu_n^{(t)} = \mu_n^{(t')} \text{ against } H_A : \mu_n^{(t)} \neq \mu_n^{(t')}, \tag{5}$$

where $\mu^{(t)}$ and $\mu^{(t')}$ are the length-$p_s$ population means in the signal subspace at timesteps $t$ and $t'$ respectively.

The oracle procedure follows these steps:

**Step 1: Untransform and extract signal dimensions**    For each observation, apply the transpose of the orthogonal matrix (its inverse, which untransforms the data) and extract the first $p_s$ dimensions:

$$Z_{nmr}^{(t)}[j] = \left( \mathcal{O}_m^\top X_{nmr}^{(t)} \right)[j], \quad j \in \{1, \ldots, p_s\}$$

**Step 2: Estimate agent-specific effects**    Estimate the agent-specific random effect $\alpha_n$ (constant across timepoints) by pooling observations across both timepoints:

$$\hat{\alpha}_n[j] = \frac{1}{2MR} \sum_{t \in \{t, t'\}} \sum_{m=1}^{M} \sum_{r=1}^{R} Z_{nmr}^{(t)}[j]$$

This is equivalent to fitting a linear mixed effects agent with agent-specific random intercepts.

**Step 3: Residualize agent effects**    Remove the estimated agent-specific effect (preserving timepoint-specific population means):

$$Y_{nmr}^{(t)}[j] = Z_{nmr}^{(t)}[j] - \hat{\alpha}_n[j]$$
$$Y_{nmr}^{(t')}[j] = Z_{nmr}^{(t')}[j] - \hat{\alpha}_n[j]$$

Afterun-rotating and residualization, $Y_{nmr}^{(t)} \approx \mu_n^{(t)} + \varepsilon_{nmr}^{(t)}$, isolating the timepoint-specific signal of interest. This follows because $O^\top \varepsilon_{nmr}^{(t)} \overset{\mathcal{D}}{=} \varepsilon_{nmr}^{(t)}$ under the isotropic measurement noise agent described in Section A.2.

**Step 4: Hotelling's $T^2$ test**    Pool residuals across all responses and replicates:

$$\mathcal{Y}^{(t)} = \{ Y_{nmr}^{(t)} : m \in \{1, \ldots, M\}, r \in \{1, \ldots, R\} \}$$
$$\mathcal{Y}^{(t')} = \{ Y_{nmr}^{(t')} : m \in \{1, \ldots, M\}, r \in \{1, \ldots, R\} \}$$

Apply the two-sample Hotelling's $T^2$ test (Hotelling, 1931; Anderson, 2003) to test $H_0 : \mu_n^{(t)} = \mu_n^{(t')}$, using the `scipy` package.

## B.3. DCorr

The `DCorr` (distance correlation) test provides a fully non-parametric alternative that makes no assumptions about transformation structure, signal dimensions, nor the functional form of temporal changes. This approach tests for distributional differences between timepoints using distance correlation (Székely et al., 2007; Székely & Rizzo, 2009), a measure of dependence between random vectors.

Under our generative agent, testing whether agent $n$ has changed between timepoints reduces to testing whether the response distributions differ for the observed queries. For the finite set of queries $Q = \{q_1, \ldots, q_M\}$, we test:

$$H_0 : f_n^{(t)}(q_m) \overset{\mathcal{D}}{=} f_n^{(t')}(q_m) \text{ for all } m \in \{1, \ldots, M\} \text{ against } H_A : f_n^{(t)}(q_m) \overset{\mathcal{D}}{\neq} f_n^{(t')}(q_m) \text{ for some } m$$

where $\overset{\mathcal{D}}{=}$ denotes equality in distribution. Note that under the generative agent (Section A.2), if $f_n^{(t)} = f_n^{(t')}$, then this null hypothesis holds for any choice of queries. The test therefore provides evidence about temporal changes in the agent's behavior on the evaluated query set.

The test proceeds by treating each response $m$ separately (to respect the independence structure of replicates), then aggregating results across responses:

**Step 1: Per-response two-sample tests**   For each response $m \in \{1, \ldots, M\}$:

(a) Pool observations from both timepoints:

$$\mathbf{X}_m = \{X_{nmr}^{(t)} : r \in \{1, \ldots, R\}\} \cup \{X_{nmr}^{(t')} : r \in \{1, \ldots, R\}\}$$

with corresponding timepoint labels $\mathbf{T}_m = [0, \ldots, 0, 1, \ldots, 1]^\top \in \{0, 1\}^{2R}$.

(b) Test independence between responses and timepoint labels using distance correlation. Under $H_0$ (no temporal change), the response distribution is independent of the timepoint label. The test statistic is:

$$\mathrm{DCorr}_m = \mathrm{DCorr}(\mathbf{X}_m, \mathbf{T}_m)$$

with p-value $p_m$ computed via permutation test, using the `hyppo` package (Panda et al., 2021).

**Step 2: Aggregate across responses**   Since response replicates are independent only within each query $q_m$, we cannot pool across responses directly. Instead, we aggregate the per-response p-values $\{p_1, \ldots, p_M\}$ using Fisher's method (Fisher, 1925):

$$\chi^2 = -2 \sum_{m=1}^{M} \log(p_m) \sim \chi_{2M}^2 \quad \text{under } H_0$$

This yields a combined test that is sensitive to temporal changes in any subset of responses.

**Key properties**   The `DCorr` approach:

- Makes no parametric assumptions about the data distribution,

- Does not require knowledge of transformation matrices or signal dimensions,

- Respects the independence structure of the replicates (independence within response, not across responses), and

- Aggregates evidence across multiple responses to detect any distributional change.

## C. Group-level tests

For a collection of agents $\{f_n^{(t)} : n \in \mathcal{G}_\ell\}$ belonging to group $\ell$ at timepoints $t$ and $t'$, we have the null hypothesis:

$$H_0 : F_\ell^{(t)} = F_\ell^{(t')} \text{ against } H_A : F_\ell^{(t)} \neq F_\ell^{(t')}, \tag{6}$$

where $F_\ell^{(t)}$ denotes the distribution of responses for agents in group $\ell$ at timepoint $t$. For each agent $n \in \mathcal{G}_\ell$, we observe independent response samples $X_{nmr}^{(t)} = g\left(f_n^{(t)}(q_m)\right)_r$ for $r \in [R]$, whose distribution depends on the agent $n$, the query $m$, and the timestep $t$.

### C.1. Paired Energy

The `PE ∘ TDKPS` group-level test leverages TDKPS embeddings to construct a fully non-parametric test that exploits the paired temporal structure of the data. This approach uses a variation of energy distance (Székely & Rizzo, 2004; 2013), a metric that detects any distributional difference between two distributions, combined with a paired permutation framework that preserves the within-agent temporal dependence structure under the null hypothesis.

Let $\{\psi_n^{(t)} : n \in \mathcal{G}_\ell\}$ be the TDKPS embeddings for all agents in group $k$ at timepoint $t$. We test whether the distribution of embeddings differs between timepoints:

$$H_0 : \psi_n^{(t)} \overset{\mathcal{D}}{=} \psi_n^{(t')} \text{ for } n \in \mathcal{G}_\ell \text{ against } H_A : \psi_n^{(t)} \overset{\mathcal{D}}{\neq} \psi_n^{(t')} \text{ for } n \in \mathcal{G}_\ell.$$

The test statistic is the energy distance between the empirical distributions at the two timepoints:

$$\delta_\ell = 2 \cdot \overline{D}_{tt'} - \overline{D}_t - \overline{D}_{t'},$$

where:

$$\overline{D}_{tt'} = \frac{1}{n_\ell(n_\ell - 1)} \sum_{\substack{n,n' \in \mathcal{G}_\ell \\ n \neq n'}} \left\| \psi_n^{(t)} - \psi_{n'}^{(t')} \right\|_2,$$

$$\overline{D}_t = \frac{1}{n_\ell(n_\ell - 1)} \sum_{\substack{n,n' \in \mathcal{G}_\ell \\ n \neq n'}} \left\| \psi_n^{(t)} - \psi_{n'}^{(t)} \right\|_2,$$

$$\overline{D}_{t'} = \frac{1}{n_\ell(n_\ell - 1)} \sum_{\substack{n,n' \in \mathcal{G}_\ell \\ n \neq n'}} \left\| \psi_n^{(t')} - \psi_{n'}^{(t')} \right\|_2,$$

and $n_\ell = |\mathcal{G}_\ell|$ is the number of agents in group $\ell$.

To assess statistical significance under the null hypothesis, we employ a paired permutation test:

**Step 1: Compute observed statistic**   Fit the TDKPS estimator to obtain embeddings $\left\{ \psi_n^{(t)} \right\}$ for all agents and timepoints. Compute the distance matrix $D$ from the embeddings, where $D_{(t,n),(t',n')} = \left\| \psi_n^{(t)} - \psi_{n'}^{(t')} \right\|_2$ for all agents $n, n'$ and timepoints $t, t'$. Compute the observed energy distance $\delta_\ell$ for group $\ell$ using the distances from $D$.

**Step 2: Generate null distribution**   For permutation $b = 1, \ldots, B$:

(a) For each agent $n \in \mathcal{G}_\ell$, randomly swap its timepoint labels with probability $0.5$, independently across agents,

(b) Compute the permuted energy distance $\delta_\ell^{(b)}$ using the same distance matrix $D$ but with the permuted timepoint assignments. Specifically, if agent $n$'s timepoints were swapped, then row/column $(t, n)$ now corresponds to agent $n$ at timepoint $t'$, and vice versa.

**Step 3: Compute $p$-value**   The p-value is computed as:

$$p = \frac{1 + \sum_{b=1}^{B} \mathbf{1}_{\left\{ \left| \delta_\ell^{(b)} \right| \geq |\delta_\ell| \right\}}}{1 + B}.$$

We use a two-tailed test (via absolute values) as energy distance can be positive or negative.

**Key properties**   The `PE ∘ TDKPS` approach:

- Exploits the paired temporal structure by permuting within agents rather than pooling observations,

- Detects any distributional difference between timepoints, not just mean shifts,

- Operates in the low-dimensional TDKPS embedding space,

- Makes no parametric assumptions about the data distribution,

- Does not require knowledge of transformation matrices nor signal dimensions, and

- Reuses the distance matrix across permutations for computational efficiency.

**Computational complexity**   The group-level test begins with the same one-time TDKPS embedding computation as the agent-level test (previously amortized across the $N$ agents), so we focus our attention on the additional terms. After extracting embeddings for the $n_\ell$ agents in group $k$ at the two timepoints, the test computes a distance matrix of size $(2 \cdot n_\ell) \times (2 \cdot n_\ell)$ in the embedding space, requiring $\mathcal{O}(n_\ell^2 \cdot d)$ operations. Critically, this distance matrix is computed once and reused across all permutations. Each of the $B$ permutations only relabels timepoint assignments within the fixed distance matrix and recomputes the energy statistic, requiring $\mathcal{O}(n_\ell^2)$ operations per permutation. Overall complexity is $\mathcal{O}((N \times T)^2 \cdot M \cdot p + n_\ell^2 \cdot d + B \cdot n_\ell^2)$, where $d \ll B$. In the worst case where $n_k = N$, the one-time kernel computation dominates and the total cost is $\mathcal{O}((N \times T)^2 \cdot M \cdot p)$, as the per-permutation cost $\mathcal{O}(B \cdot N^2)$ is negligible in comparison.

On the other hand, $N$ agent-level tests require $\mathcal{O}(B \cdot N^2 \cdot T \cdot M \cdot p)$ or $\mathcal{O}(B \cdot N^3 \cdot T^2 \cdot d)$ operations depending on the regime, in addition to the one-time amortized TDKPS embedding computation. $BN^2$ is dominated by many, many orders of magnitude by either of these terms as in particular, $N$, $M$, and $p$ will typically be quite large. This efficiency gain arises from subverting kernel updates and re-embeddings for each permutation, instead reusing a single distance matrix with simple relabeling operations. Further, the one-time amortized TDKPS embedding cost is dominated by these terms whenever $B \gg T$, which is likely to also often be the case.

## C.2. Oracle

The `oracle` group-level test represents an upper bound on performance for detecting mean shifts by leveraging knowledge unavailable in practice. This oracle knows:

1. the true orthogonal matrices $\mathcal{O}_m$ for each response $m$,

2. that signal is concentrated in the first $p_s$ dimensions after untransformation, and

3. the generative agent structure (population means plus agent-specific random effects plus noise) to extract the optimal signal from each agent.

Under the generative agent, testing Equation (6) for mean shifts is equivalent to testing whether the population means in the signal subspace differ between timepoints for group $\ell$:

$$H_0 : \mu_\ell^{(t)} = \mu_\ell^{(t')} \text{ against } H_A : \mu_\ell^{(t)} \neq \mu_\ell^{(t')}, \tag{7}$$

where $\mu_\ell^{(t)}$ is the length-$p_s$ population mean for group $\ell$ in the signal subspace at timepoint $t$.

The `oracle` procedure follows these steps:

**Step 1: Untransform and extract signal dimensions**   For each agent $n \in \mathcal{G}_\ell$ and observation, apply the transpose of the orthogonal matrix and extract the first $p_s$ dimensions:

$$Z_{nmr}^{(t)}[j] = \left( \mathcal{O}_m^\top X_{nmr}^{(t)} \right)[j], \quad j \in \{1, \ldots, p_s\}$$

**Step 2: Average over responses and replicates**   For each agent $n \in \mathcal{G}_\ell$, compute the averaged untransformed signal vector at each timepoint:

$$\bar{Z}_n^{(t)}[j] = \frac{1}{MR} \sum_{m=1}^{M} \sum_{r=1}^{R} Z_{nmr}^{(t)}[j]$$

This gives $\bar{Z}_n^{(t)} \in \mathbb{R}^{p_s}$ for each agent at each timepoint. Note that due to the fact that responses (after untransformation) are identically distributed within a given agent, this is the optimal estimate of the agent-specific mean vector (including agent-specific random effects).

**Step 3: Compute within-agent differences**   For each agent $n \in \mathcal{G}_\ell$, compute the temporal difference:

$$D_n = \bar{Z}_n^{(t')} - \bar{Z}_n^{(t)} \in \mathbb{R}^{p_s}$$

Due to the paired structure, this also residualizes the agent-specific random effects from the agent. Under $H_0$, $\mathbb{E}[D_n] = 0$ since the population means are equal across timepoints.

**Step 4: Paired Hotelling's $T^2$ test**   Apply the paired Hotelling's $T^2$ test (Hotelling, 1931; Anderson, 2003) to test $H_0 : \mathbb{E}[D_n] = 0$:

$$\bar{D} = \frac{1}{n_\ell} \sum_{n \in \mathcal{G}_\ell} D_n,$$

$$S_D = \frac{1}{n_\ell - 1} \sum_{n \in \mathcal{G}_\ell} (D_n - \bar{D})(D_n - \bar{D})^\top,$$

$$T^2 = n_\ell \cdot \bar{D}^\top S_D^{-1} \bar{D}.$$

The test statistic is transformed to an F-statistic:

$$F = \frac{n_\ell - p_s}{(n_\ell - 1)p_s} T^2 \sim F_{p_s, n_\ell - p_s} \quad \text{under } H_0.$$

**Key properties**   The `oracle` approach:

- Provides a near-optimal upper bound for detecting mean shifts,

- Exploits the paired temporal structure through paired differences $D_n$,

- Removes nuisance dimensions by projecting to the signal subspace,

- Averages over responses to reduce noise, and

- Requires knowledge of the true generative agent, unavailable in practice.

### C.3. DCorr

The `DCorr` (distance correlation) baseline provides an unpaired, fully non-parametric group-level test that operates directly in the high-dimensional ambient response space. Unlike the paired energy test, this approach treats observations from the two timepoints as independent samples and tests for distributional differences using a two-sample framework.

For a group $\mathcal{G}_\ell \subseteq [N]$ of agents, we test:

$$H_0 : F_\ell^{(t)} = F_\ell^{(t')} \text{ against } H_A : F_\ell^{(t)} \neq F_\ell^{(t')},$$

where $F_\ell^{(t)}$ denotes the distribution of responses for agents in group $k$ at timepoint $t$.

The test proceeds as follows:

**Step 1: Average over replicates**   For each agent $n \in \mathcal{G}_\ell$, compute the averaged response vector at each timepoint:

$$\bar{X}_n^{(t)} = \frac{1}{R} \sum_{r=1}^R X_{nmr}^{(t)} \in \mathbb{R}^{M \times p}$$

where $M$ is the number of responses and $p$ is the feature dimension. Flatten this into a vector:

$$\tilde{X}_n^{(t)} = \text{vec}(\bar{X}_n^{(t)}) \in \mathbb{R}^{Mp}$$

**Step 2: Pool observations across timepoints**   Create a pooled data matrix by stacking observations from both timepoints:

$$X_\ell = \begin{bmatrix} \tilde{X}_{n_1}^{(t)} \\ \vdots \\ \tilde{X}_{n_{n_\ell}}^{(t)} \\ \tilde{X}_{n_1}^{(t')} \\ \vdots \\ \tilde{X}_{n_{n_\ell}}^{(t')} \end{bmatrix} \in \mathbb{R}^{2n_\ell \times Mp}$$

with corresponding timepoint labels:

$$Y_\ell = [0_{n_\ell}, 1_{n_\ell}]^\top \in \{0, 1\}^{2n_\ell}$$

**Step 3: Test independence via distance correlation**   Apply the distance correlation test (Székely et al., 2007; Székely & Rizzo, 2009) to test whether the response vectors $X_\ell$ are independent of the timepoint labels $Y_\ell$. Under $H_0$ (no temporal change), responses should be independent of timepoint membership (Panda et al., 2025). The test statistic is:

$$\text{DCorr}_\ell = \text{DCorr}(X_\ell, Y_\ell)$$

with p-value computed via permutation test (randomly permuting the timepoint labels), using the `hyppo` package (Panda et al., 2021).

**Key properties**   The `DCorr` approach:

- Makes no parametric assumptions about the data distribution,

- Detects any form of distributional difference between timepoints,

- Operates in the high-dimensional ambient space ($Mp$ dimensions),

- Treats observations as independent samples, ignoring the paired temporal structure,

- Does not require knowledge of transformation matrices nor signal dimensions, and

- May suffer from the curse of dimensionality when $Mp$ is large relative to $n_\ell$.

### C.4. DCorr ∘ TDKPS

The `DCorr ∘ TDKPS` hybrid baseline combines TDKPS dimensionality reduction with the unpaired distance correlation testing framework. This approach benefits from operating in a lower-dimensional embedding space while still treating the two timepoints as independent samples.

Let $\{\psi_n^{(t)} : n \in \mathcal{G}_\ell\}$ be the TDKPS embeddings for all agents in group $\ell$ at timepoint $t$, where $\psi_n^{(t)} \in \mathbb{R}^d$ with $d \ll Mp$. Further, assume that each $\psi_n^{(t)}$ are independent samples from $G_\ell^{(t)}$. We seek to test whether:

$$H_0 : G_\ell^{(t)} = G_\ell^{(t')} \text{ against } H_A : G_\ell^{(t)} \neq G_\ell^{(t')},$$

which is a 2-sample test (Panda et al., 2025).

The test proceeds as follows:

**Step 1: Obtain TDKPS embeddings**   Fit the TDKPS estimator to the full dataset to obtain embeddings $\{\psi_n^{(t)} : n \in [N], t \in [T]\}$. Extract embeddings for group $\ell$ at both timepoints.

**Step 2: Pool embeddings across timepoints**   Create a pooled embedding matrix by stacking embeddings from both timepoints:

$$\Psi_\ell = \begin{bmatrix} \psi_{n_1}^{(t)} \\ \vdots \\ \psi_{n_{n_\ell}}^{(t)} \\ \psi_{n_1}^{(t')} \\ \vdots \\ \psi_{n_{n_\ell}}^{(t')} \end{bmatrix} \in \mathbb{R}^{2n_\ell \times d}$$

with corresponding timepoint labels:

$$Y_\ell = [0_{n_\ell}, 1_{n_\ell}]^\top \in \{0, 1\}^{2n_\ell}$$

**Step 3: Test independence via distance correlation**  Apply the distance correlation test to test whether the TDKPS embeddings $\Psi_\ell$ are independent of the timepoint labels $Y_\ell$:

$$\text{DCorr}_\ell = \text{DCorr}(\Psi_\ell, Y_\ell)$$

with $p$-value computed via permutation test, using the `hyppo` package.

**Key properties**  The `DCorr ∘ TDKPS` approach:

- Operates in the low-dimensional TDKPS embedding space ($d$ dimensions, where $d \ll Mp$),

- Makes no parametric assumptions about the data distribution,

- Benefits from TDKPS dimensionality reduction and denoising,

- Treats observations as independent samples, ignoring the paired temporal structure,

- Does not require knowledge of transformation matrices nor signal dimensions, and

- Conceptually allows isolation of the benefit of dimensionality reduction versus exploiting paired structure when compared to `DCorr` and `PE ∘ TDKPS`.

## D. Sensitivity to Pipeline Choices

A practical concern for any monitoring framework is robustness to choices in the data processing pipeline. We investigate sensitivity to two key choices in our congressperson system: the text embedding model $g$ and the number of retrieved tweets per query.

### D.1. Embedding Model

We replicate the group-level analysis (Section 3.3) using four open-source embedding models spanning different training pipelines, architectures, and embedding dimensions (Table 4). For each model, we recompute the full TDKPS pipeline and group-level test statistics for all consecutive timepoint pairs on the public health query set with retrieval depth fixed at 2.

*Table 4.* Embedding models used in the sensitivity analysis.

| Model | Dimension | Source |
|---|---|---|
| `nomic-embed-text-v1.5` | 768 | Nomic AI |
| `all-MiniLM-L6-v2` | 384 | Sentence-Transformers |
| `gte-large` | 1024 | Alibaba DAMO |
| `bge-large-en-v1.5` | 1024 | BAAI |

To measure agreement, we compute pairwise Spearman rank correlations between the vectors of group-level test statistic magnitudes across all 13 consecutive timepoint pairs. Table 5 reports the results.

*Table 5.* Pairwise Spearman rank correlations of group-level test statistic magnitudes across embedding models (retrieval depth $= 2$).

| | nomic | minilm | gte | bge |
|---|---|---|---|---|
| `nomic` | 1.000 | 0.996 | 0.998 | 0.992 |
| `minilm` | | 1.000 | 0.997 | 0.995 |
| `gte` | | | 1.000 | 0.997 |
| `bge` | | | | 1.000 |

All pairwise correlations exceed 0.99, indicating that the temporal pattern of detected behavioral shifts is essentially invariant to the choice of embedding model across the architectures and training procedures considered.

## D.2. Retrieval Depth

We vary the number of retrieved tweets per query ($n_{\text{tweets}} \in \{1, 2, 3\}$) while holding the embedding model fixed at `nomic-embed-text-v1.5`. Table 6 reports pairwise Spearman rank correlations of group-level test statistic magnitudes.

*Table 6.* Pairwise Spearman rank correlations of group-level test statistic magnitudes across retrieval depths (embedding model = `nomic-embed-text-v1.5`).

| $n_{\text{tweets}}$ | 1 | 2 | 3 |
|---|---|---|---|
| 1 | 1.000 | 0.812 | 0.831 |
| 2 | | 1.000 | 0.874 |
| 3 | | | 1.000 |

Correlations across retrieval depths are strong (0.81–0.87), though lower than across embedding models. This is expected: varying retrieval depth changes the *input* to each agent (the retrieved context), which can genuinely alter agent behavior, whereas varying the embedding model changes only the *observation function* applied to a fixed set of responses. The high but imperfect correlations suggest that while the broad temporal pattern is robust to retrieval depth, fine-grained test statistic magnitudes are moderately sensitive to the amount of retrieved context.

## D.3. Retrieval Quality

A potential concern with our congressperson system is whether the top-2 retrieved tweets actually contain information relevant to the query, or whether the agent is effectively hallucinating from uninformative context. To assess this, we compute, for each (congressperson, query) pair, the average cosine similarity between the query embedding and the two retrieved tweet embeddings, and compare it to the average cosine similarity between the query embedding and a randomly selected tweet from the same congressperson's history. We report the median and interquartile range (IQR) of the paired difference (retrieved similarity minus random similarity) across congresspersons in Table 7.

*Table 7.* Retrieval quality: paired difference in cosine similarity between retrieved tweets and a random tweet baseline, summarized across congresspersons. A positive difference indicates that the retrieval mechanism selects tweets more relevant to the query than a random baseline.

| Topic | Avg. random sim. | Median $\Delta$ | 25% $\Delta$ | 75% $\Delta$ |
|---|---|---|---|---|
| Public health | 0.480 | 0.105 | 0.089 | 0.122 |
| General politics | 0.451 | 0.154 | 0.131 | 0.181 |
| Orthogonal (null) | 0.462 | 0.089 | 0.076 | 0.104 |

The paired difference is strictly positive for every (congressperson, query) pair across all three topic conditions, confirming that the retrieval mechanism consistently grounds the agent in the most relevant available context. The effect is largest for general political queries (median $\Delta = 0.154$), where congresspersons' tweet histories tend to contain abundant directly relevant content, and smallest for orthogonal queries (median $\Delta = 0.089$), where relevant tweets are naturally scarcer. Importantly, even for the orthogonal (candy & chocolate) queries, retrieval selects measurably more relevant context than a random baseline, indicating that the retrieval mechanism functions as intended across all topic conditions.

## D.4. Representative Query–Response Examples

To illustrate the quality of retrieved context and the resulting agent responses, we select the congressperson whose average cosine similarity across all public health queries is closest to the overall median: Brendan F. Boyle (D-PA) (average cosine similarity = 0.590). We display representative (query, retrieved tweets, response) triplets at the 2020-04-01 timepoint. Additional examples for all three query sets, spanning a range of retrieval similarities, are provided in the supplementary materials.

PUBLIC HEALTH EXAMPLE (MINIMUM COSINE SIMILARITY = 0.527)

**Query:** *How did perceptions of risk influence compliance with quarantine measures?*

**Retrieved Tweet 1:** "This is an excellent chart. We must learn from the painful lessons of my hometown's experience a century ago, when they carried on with a public parade that helped infect tens of thousands. Practice #SocialDistancing"

**Response:** "Based on my experience and personal connections with communities across Pennsylvania, perceptions of risk played a significant factor. Personal connections—seeing COVID-19 take a toll or tragically take lives in one's community—swayed many citizens towards heightened adherence to public health guidelines..."

Even at the minimum retrieval similarity, the retrieved tweets provide tangential but relevant historical context that the agent leverages to produce a persona-consistent response grounded in the congressperson's public communication style. This pattern holds broadly across congresspersons and query sets: retrieved tweets supply topically relevant anchoring information, and the agent integrates this context into responses that reflect the congressperson's documented positions and rhetorical style.

D.4.1. NOTE ON QUERY SET CONTENT

The full set of 100 queries per topic, generated via the prompts in Appendix A.1, covers a range of subtopics designed to probe different facets of each domain. For public health, subtopics include health communication and misinformation, science and society, risk perception and public engagement, and health policy and governance. For general politics: immigration, the economy, LGBTQ rights, and climate change. For the orthogonal condition: chocolate, frozen treats, taffy and toffee, and fruit-flavored gummies.

