# OpenReview forum: "Detecting Perspective Shifts in Multi-Agent Systems"
_ICML.cc/2026/Conference — ICML 2026 regular_

### Official Review · Reviewer_tGy2 · 2026-03-10

**Soundness:** 3
**Presentation:** 3
**Significance:** 3
**Originality:** 3
**Overall Recommendation:** 3
**Confidence:** 3

**Summary:**

The paper introduces the Temporal Data Kernel Perspective Space for detecting behavioral changes (perspective shifts) in black-box multi-agent systems over time. It represents agents by their responses to a reference set of queries and jointly embeds these representations across multiple time points into a low-dimensional Euclidean space. It also propose several hypothesis tests for detecting shifts at both the individual agent level and group level. The framework's effectiveness is demonstrated through simulations and a "natural experiment" involving digital congresspersons, where the tests successfully detected significant perspective shifts in public health discourse following the onset of the COVID-19 pandemic.

**Compliance With Llm Reviewing Policy:**

Affirmed.

**Final Justification:**

My final recommendation remains unchanged after considering the paper and the rebuttal. The rebuttal only partially resolves my concerns so that and I maintain my score.

**Key Questions For Authors:**

Q1.You noted that Type-I error inflation decreases as the number of response replicates (R) increases. In scenarios where R is limited by budget or latency, are there other calibration techniques you would suggest to stabilize the null distribution?

Q2.How could the framework be modified to handle agents with internal state or memory, where the independence assumption between query-response pairs is violated?

Q3.Given the sensitivity to the query set, have you considered using automated methods (e.g., LLM-based red-teaming or diversity-seeking algorithms) to generate the reference query set Q?

**Limitations:**

yes

**Strengths And Weaknesses:**

**Strength**
S1. The paper proposes a framework for monitoring behavioral dynamics in black-box multi-agent systems, filling a critical gap in the literature where internal mechanism access is often assumed but unavailable.

S2. The "natural experiment" using digital congresspersons provides strong empirical evidence, demonstrating that the method can detect meaningful shifts tied to real-world exogenous events like the COVID-19 pandemic.

S3. In simulations, the framework consistently achieves statistical power close to an "oracle" baseline, proving that its low-dimensional embeddings retain most of the relevant signal for shift detection. The proposed group-level test is computationally efficient and scales well.

**Weakness**

W1. The agent-level test exhibits a slight inflation of Type-I error (rejection rates of 5-10% for the null class), which might lead to false positives in high-stakes monitoring scenarios.

W2. The effectiveness of the framework is highly dependent on the choice of the query set Q, yet the paper provides limited formal guidance on how to construct a "sufficiently expansive" or "optimal" set of queries.

W3. The current approach assumes that query-response pairs are independent samples, which may not adequately capture the behavior of agents that maintain conversational state or long-term dialogue history.

---

> ### Author Rebuttal · Authors · 2026-03-27
>
> Thank you for taking the time to provide a thorough review. We think your summary is a fair representation of the work. We also appreciate the acknowledgement that the work fills a critical gap in the current literature and that the method is effective at detecting real-world exogenous. We address the mentioned weaknesses / questions by number and short description.
>
> > Q1. Calibration with limited budget / memory
>
> The calibration issue for small $ R $ was mentioned as a weakness by the majority of reviewers. Given this, we designed a better-calibrated test and profiled its Type-I error and power in a selected setting in our response to Reviewer pLsC. We will include a full profile of the calibrated test in the agent-level simulation figure.
>
> > Q2. Handling changes in internal state
>
> Generally, the TDKPS framework naturally handles agents with changing internal states / memories. With that said, the hypothesis test we propose addresses the “zeroth order” change point problem and is sensitive to changes in the internal state that affect the agent’s behavior with respect to the query set.
>
> In settings where the operator wants to use a monitoring system that allows changes in the internal state that produce "benevolent" changes in the agent’s behavior, a “first order” detection method is necessary. First order detection systems require a burn-in period to establish what “benevolent” changes look like. We think that this is a very important next-phase of research in the TDKPS lineage. We will add an additional sentence in the discussion / limitations section to explicitly mention this extension.
>
>
> > Q3. Generating a query set
>
> As demonstrated in our real-data analysis, the query set is a first-class citizen in TKDPS-based monitoring systems – and likely any black-box monitoring system. In short, yes we have considered / implemented automatically generated query sets. In particular, people who use platforms that implement TDKPS-based methods typically have bespoke monitoring goals. The quality of query sets and its effect on black-box inference is a relatively new but important topic of generative model / agent research – practitioners must be sure that the query set is relevant, comprehensive, and simultaneously not-too-small (or it is too sensitive) and not-too-large (or it is too expensive). While we do not have more rigorous guidelines at this time, we will add a sentence to our discussion highlighting these challenges more in-depth.

---

> > ### Author Rebuttal · Reviewer_tGy2 · 2026-04-01
> >
> > Thank you for the rebuttal.  The first concern is solved but the other two concerns are not. I will keep my score unchanged at this point.

---

### Official Review · Reviewer_3Ujy · 2026-03-11

**Soundness:** 3
**Presentation:** 3
**Significance:** 3
**Originality:** 3
**Overall Recommendation:** 5
**Confidence:** 3

**Summary:**

The paper introduces the Temporal Data Kernel Perspective Space (TDKPS), a statistical framework designed to detect behavioral changes in multi-agent systems. The authors propose agent-level and group-level hypothesis tests based on embedding the agents' responses to fixed query sets. The methodology is evaluated on simulated data and a real-world "natural experiment" consisting of 99 LLM-driven "digital congresspersons" conditioned on historical tweets. The framework successfully identifies significant behavioral shifts in public health queries that align temporally and specifically with the onset of the COVID-19 pandemic, demonstrating its utility for monitoring black-box generative systems.

**Compliance With Llm Reviewing Policy:**

Affirmed.

**Key Questions For Authors:**

1. The current experiments are limited in two ways: (1) the Gaussian blob simulations are highly idealized, and (2) the real-world setup evaluates isolated RAG pipelines rather than interacting agents. Could you evaluate TDKPS on a true LLM-based MAS dataset (e.g., an LLM debate, open-source generative agent sandbox, or generative agent responses in a social simulation) where agents actually interact and influence each other? Providing a demonstration on natively interacting agents would directly bridge the gap between the paper’s framing and its empirical evidence.

2. The reliance on a single 8B model from 2024 raises questions about broader applicability. Could you run an ablation using a different, preferably larger or more modern model (e.g., Gemini 3 Flash, GPT-5-mini)? If API cost is a concern, consider running this on a smaller scale.

3. Since responses rely heavily on the top 2 retrieved tweets, how did you ensure these tweets were actually relevant to the 100 specific topic questions? Can you provide an analysis (e.g., average cosine similarity between queries and retrieved context) to prove the LLM was grounded rather than hallucinating? Confirming that the agents were responding based on relevant historical context rather than hallucinating due to missing data is critical to validating the design of the "natural experiment."

4. Can you provide a representative sample of the 100 queries and their corresponding LLM responses (e.g., in the rebuttal and a revised appendix)? Additionally, are there plans to release the code and dataset for reproducibility?

If the authors disagree with any of the points or critiques raised, please clarify. I would love to engage in discussion and am open to increasing my score if my main concerns are adequately addressed.

**Limitations:**

yes

**Strengths And Weaknesses:**

##  Strengths

1. Detecting LLM-agent behavioral change is a crucial and urgent problem regarding the safety and reliable deployment of multi-agent systems (MAS) in the real world.

2. The methodology operates entirely on system inputs and outputs. By removing the need for access to model weights, gradients, or internal states, it is highly applicable to proprietary commercial models.

3. The introduction of the group-level test (PE ◦ TDKPS) effectively solves the computational bottleneck of the agent-level test, allowing the framework to scale to larger systems.

4. The statistical analysis is rigorous, multifaceted, and well-supported by thorough simulation studies.

## Weaknesses

1. The paper frames its contribution around "dynamic multi-agent systems," but the primary empirical validation consists of 99 entirely isolated, independent Retrieval-Augmented Generation (RAG) pipelines. There is no actual multi-agent interaction, autonomous decision-making, feedback loops, tool use, or shared environment evaluated, making the framing somewhat misleading.
- The experiment feels more like detecting in-situ response embedding changes conditioned on real-world Twitter data, which does not capture the complex interactions of actual language agents evolving over time. LLM-based agent systems can exhibit behavioral distribution shifts that are drastically different from real-world humans due to issues such as limited agent memory, context drift, and hallucinations. Treating real-world human Twitter opinion data as a simulated proxy may not be a direct empirical measurement.
- The paper would greatly benefit from evaluating actual data generated natively from multiple LLMs interacting with each other, as seen in recent LLM-based social simulations.
2. Fundamentally, the method detects a distribution shift in language embedding (naming “Perspective Shifts”), complicated by inter-agent and inter-topic variance. TDKPS appears to be an application of existing dimensionality reduction techniques (MDS) combined with additional permutation statistical tests, rather than a novel "first-principles framework." It is difficult to judge the true algorithmic or theoretical novelty based on the current writing.
3. The simulation setup relies on contrived Gaussian blobs with rigid, pre-defined signal decay and noise parameters. While the authors ablate sample sizes and effect magnitudes, this idealized parametric data generation process does not reflect the complex, non-linear linguistic shifts of actual LLMs. Consequently, there remains a significant sim2real gap, making it difficult to trust that the high statistical power observed in simulation will translate to complex generative text distributions.
4. The paper only tested a single model, Ministral-8B-Instruct-2410. Real-world LLM-agent systems mostly use commercial APIs with much larger and more capable models. An 8B model from 2024’s behavioral distribution, reasoning capability, and propensity to hallucinate could be drastically different from state-of-the-art models deployed in practice.
5. The query setup consists of 100 topic questions intended to simulate a real-world congressperson's views by retrieving the 2 most relevant tweets as conditioning. However, there is no discussion or "coverage" analysis regarding whether the retrieved tweets actually contain relevant information about the topics.
- Consider the extreme case where a congressperson never tweeted anything relevant about any listed topics; the LLM's response would be entirely hallucinated.
6. No concrete examples of the 100 topic questions or the resulting LLM responses are provided in the main text or appendix. This omission makes it difficult to judge the complexity, length, diversity, and richness of the queries and responses, further limiting the judgment of soundness, reproducibility, and qualitative assessment.

---

> ### Author Rebuttal · Authors · 2026-03-27
>
> Thank you for taking the time to provide a thorough and detailed review. We think your summary is a fair representation of our work, and we appreciate your acknowledgement that the problem is crucial and urgent, that our framework / approach is highly applicable, and that our analysis is rigorous. We address your questions and some of the mentioned weaknesses below.
>
> > 1a. Idealized simulations
>
> While the Gaussian blob simulations are idealized, the controlled parametric setup allows us to cheaply and precisely study the effect of many relevant parameters (effect size, number of agents, number of queries, number of replicates) in a way that would be intractable with LLM-based MAS dataset / social simulation. Reviewer uP9F suggested studying the effect of "sparse" changes in the simulation setting to better mimic LLM-based MAS simulations. Please see our response to them for that analysis.
>
> > 1b. Isolated RAG pipelines
>
> We disagree that the agents in our experiment are entirely isolated. The dynamic process that determines the contents of each congressperson's tweet history innately involves congresspersons interacting with each other (they respond to each other publicly, interact with shared legislation, etc.). While this interaction graph is complicated and latent, the observed data (the Tweets) is dependent on it. Further, this setting (where the researcher / operator does not have access to the interaction graph or internal states of the underlying dynamics) is precisely the black-box setting that motivates our change point detection / monitoring systems. We will clarify this framing in the main text.
>
> > 2. Reliance on a single model
>
> This is a valid weakness. Our computation setup relies on consumer-grade GPUs rather than commercial APIs, which limits our ability to evaluate larger or more recent models. With that said, in our response to Reviewer uP9F we showed that the group-level test statistics are highly robust, with nearly perfect ranked correlations, across four embedding models spanning different architectures, training pipelines, and embedding dimensions and across multiple retrieval depths. While this does not directly address the choice of generative model, it suggests the framework's conclusions are not brittle to some pipeline choices. We will add the single-model limitation to the discussion.
>
> > 3. Retrieval quality / hallucination concern
>
> We did not evaluate the groundedness in our analysis initially. We have since computed the average paired difference in cosine similarity between the retrieved tweets and a random tweet. We report the median and IQR (across congresspersons) in the table below.
>
> |Topic|Avg. random sim.|Median|25%|75%|
> |---|---:|---:|---:|---:|
> |Public health|0.480|0.105|0.089|0.122|
> |General politics|0.451|0.154|0.131|0.181|
> |Orthogonal (null)|0.462|0.089|0.076|0.104|
>
> The average paired difference is strictly positive for every (congressperson, query) pair across all three topic conditions. This confirms that the retrieval mechanism grounds the LLM in the most relevant available context rather than hallucinating from an empty conditioning set. We will include this analysis in a new appendix section, which will contain these diagnostic-level plots as well as our sensitivity analyses.
>
> > 4. Representative samples
>
> To illustrate retrieval quality concretely, we select the congressperson whose average cosine similarity across all public-health queries is closest to the median: Brendan F. Boyle (D-PA) (avg. cosine sim=0.590). We display (query, retrieved tweets, response) triplets at the min of query-tweet cosine similarity, all at the 2020-04-01 timepoint.
>
> Min (cosine sim=0.527)
> Query: How did perceptions of risk influence compliance with quarantine measures?
> Tweet 1: "This is an excellent chart. We must learn from the painful lessons of my hometown's experience a century ago, when they carried on with a public parade that helped infect tens of thousands. Practice #SocialDistancing"
> Response: "Based on my experience and personal connections with communities across Pennsylvania, perceptions of risk played a significant factor. Personal connections — seeing COVID-19 take a toll or tragically take lives in one's community — swayed many citizens towards heightened adherence to public health guidelines..."
>
> Even at the minimum cosine similarity, retrieved tweets provide tangential but relevant context that the model leverages to produce a persona-consistent response. While we cannot provide more examples due to rebuttal length constraints, representative samples (query, retrieved tweets, response) for each of the three collections of queries will be included in the appendix.
>
> > 5. Code and data release
>
> Yes. All code, data, and analysis for every figure will further be made publicly available on Github. The code will further be available via a pip installable Python package.
>
> Thank you again for your thorough review and for your openness to further discussion.

---

> > ### Author Rebuttal · Reviewer_3Ujy · 2026-04-01
> >
> > My main concerns regarding the experimental setup have been addressed, and I have adjusted my score accordingly. Thank you to the authors for the substantial additional experiments and clarifications.

---

### Official Review · Reviewer_uP9F · 2026-03-13

**Soundness:** 3
**Presentation:** 3
**Significance:** 2
**Originality:** 2
**Overall Recommendation:** 5
**Confidence:** 3

**Summary:**

This paper introduces the Temporal Data Kernel Perspective Space (TDKPS), a framework for embedding black-box agents across time into a low-dimensional Euclidean space, and proposes hypothesis tests for detecting behavioral change at both agent-level and group-level. TDKPS extends the static Data Kernel Perspective Space (DKPS) of Helm et al. (2024) to the temporal setting by jointly embedding agents across multiple timepoints via classical multidimensional scaling on a block distance matrix constructed from averaged embedded query responses. The paper proposes two tests: an agent-level test based on the Euclidean distance between an agent's TDKPS representations at two timepoints with permutation-based significance, and a group-level test (PE∘TDKPS) using an energy distance statistic with paired permutations. In simulations with temporal Gaussian blobs, the tests achieve near-oracle power while substantially outperforming a DCorr baseline. The real-data application constructs 99 digital congresspersons (Ministral-8B agents with congressperson-specific tweet databases) over 14 timepoints (2018–2024) and demonstrates that behavioral shifts detected by TDKPS on public health queries correlate specifically with the onset of COVID-19 (Kendall's tau = 0.51, p = 0.014), while orthogonal (candy) queries show no such pattern (K = 0.01, p = 0.956).

**Compliance With Llm Reviewing Policy:**

Affirmed.

**Final Justification:**

My concerns have been adequately addressed. I believe this is an valuable idea (using kernel embeddings to detect agent behaviors) and would like to slightly raise my score.

**Key Questions For Authors:**

1. **How does TDKPS perform when behavioral changes are non-linear or sparse (e.g., an agent changes its response to 5% of queries dramatically)?** The simulation model assumes a shift in class means across a signal subspace, which is well-suited to the Euclidean/MDS framework. A simulation with more realistic LLM-like behavioral changes (e.g., categorical response flips on a subset of queries) would clarify whether TDKPS's power extends beyond the Gaussian setting.

2. **Can you provide a principled correction for the Type-I error inflation in the agent-level test, or at minimum, characterize it analytically as a function of N, T, M, R?** Without valid p-values, the agent-level test is difficult to use in practice. If correction is intractable, would a bootstrap approach that jointly resamples agents and replicates provide better calibration?

3. **How sensitive are the real-data conclusions to the choice of embedding model (nomic-embed-v1.5) and the number of retrieved tweets (top-2)?** If TDKPS is proposed as a general monitoring framework, understanding its robustness to these pipeline choices is essential. A brief ablation varying the embedding model or retrieval depth would help.

**Limitations:**

yes

**Strengths And Weaknesses:**

**Strengths:**
- The paper addresses a genuinely important and timely problem: principled statistical monitoring of behavioral dynamics in black-box multi-agent systems. The formulation is clean — TDKPS reduces complex agent dynamics to a tractable geometric problem in Euclidean space, and the proposed permutation tests are well-designed for the paired temporal structure. The group-level test's computational efficiency (O(N²) per permutation vs. full re-embedding) is a practical advantage.
- The real-data experiment is thoughtfully designed as a natural experiment. The use of three query sets (public health, general politics, candy) as differential controls is a strong methodological choice that allows the authors to distinguish COVID-19-specific behavioral shifts from generic temporal drift. The Kendall's tau analysis (Figure 4III) provides convincing evidence that detected shifts are temporally concentrated around COVID-19 for public health queries specifically.
- The paper is honest about its limitations. The authors acknowledge Type-I error inflation (5–10%) for the agent-level test at R=25, the computational trade-off they made (accepting inflated Type-I over 4× cost), query set dependence, and the inability to handle context-dependent agent behaviors. This transparency strengthens the contribution.

**Weaknesses:**
- The technical novelty relative to prior DKPS work (Helm et al., 2024; Acharyya et al., 2025) is incremental. TDKPS is essentially DKPS applied to a block distance matrix that concatenates agents across timepoints, solved by standard classical MDS (Eq. 1). The hypothesis tests are standard permutation tests on Euclidean distances or energy distances. The conceptual contribution — applying DKPS temporally and proposing hypothesis tests — is valuable, but the methodological depth is limited for ICML.
- The simulation model (temporal Gaussian blobs) is highly stylized and may overstate the method's effectiveness. The signal structure (front-loaded to back-loaded shift in a known subspace, observed through random orthogonal transformations plus Gaussian noise) is perfectly aligned with the Euclidean distance metric and MDS embedding that TDKPS uses. It is unclear how TDKPS would perform under non-Gaussian, non-linear, or sparse behavioral changes that are more realistic for LLM-based agents.
- The agent-level permutation test has a fundamental structural issue acknowledged but not resolved: permuting one agent's replicates alters the shared distance matrix and thus affects all agents' embeddings, leading to Type-I error inflation. This means the agent-level test cannot produce valid p-values at the nominal level without large R, which significantly limits its practical utility for monitoring deployed systems where collecting many replicates per query may be expensive.

---

> ### Author Rebuttal · Authors · 2026-03-27
>
> Thank you for taking the time to review our paper. We think your summary is a fair representation of our work. We also appreciate your comments on the timeliness / importance of the problem we address and the design of our real-data experiment. We address your mentioned weaknesses by number.
>
> > 1.
>
> Great question. There are two regimes worth considering here: i) where the behavior of the agent changes on a small amount of queries but the total amount of signal is preserved and ii) where the behavior of the agent changes on a small amount of queries and the total amount of signals scales with the number of queries. We provide initial results estimated from 50 Monte Carlo replicates with 200 permutations per test and $\alpha=0.05$ below. $\pm$ denotes standard error. Both the uncalibrated and calibrated (sample-split) tests are reported.
>
> ## Scaling Signal
>
> Per-query shift fixed at $\delta=1.0$; total signal grows with proportion of changed queries $s$.
>
> |$s$|$\delta$|C0 Power (Uncalib)|C0 Power (Calib)|C1 Type 1 (Uncalib)|C1 Type 1 (Calib)|
> |---|---:|---:|---:|---:|---:|
> |0.05|1.00|0.083±0.013|0.057±0.011|0.073±0.010|0.039±0.009|
> |0.10|1.00|0.116±0.018|0.084±0.015|0.074±0.013|0.048±0.011|
> |0.25|1.00|0.192±0.028|0.123±0.016|0.054±0.011|0.055±0.008|
> |0.50|1.00|0.584±0.039|0.390±0.036|0.070±0.013|0.061±0.010|
> |1.00|1.00|0.928±0.019|0.788±0.033|0.050±0.011|0.029±0.007|
>
> When the total signal scales with sparsity, power increases monotonically with $s$. At $s=0.05$ (1 of 20 queries changed), the signal is weak and barely above the nominal level for the uncalibrated test and not above it for the calibrated test.
>
> ## Constant Signal
>
> Total signal energy fixed; per-query shift scales as $\delta/\sqrt{s}$.
>
> |$s$|$\delta$|C0 Power (Uncalib)|C0 Power (Calib)|C1 Type 1 (Uncalib)|C1 Type 1 (Calib)|
> |---|---:|---:|---:|---:|---:|
> |0.05|4.47|0.924±0.020|0.798±0.031|0.071±0.009|0.036±0.007|
> |0.10|3.16|0.918±0.018|0.779±0.034|0.071±0.011|0.053±0.012|
> |0.25|2.00|0.871±0.025|0.696±0.040|0.042±0.010|0.047±0.008|
> |0.50|1.41|0.915±0.023|0.802±0.034|0.067±0.012|0.054±0.010|
> |1.00|1.00|0.923±0.021|0.781±0.032|0.056±0.011|0.032±0.008|
>
> When total signal is held constant, power is approximately flat across all sparsity levels (0.70–0.80 for the calibrated test, 0.87–0.92 uncalibrated). That is, the agent-level test detects the temporal change equally well whether the signal is concentrated on a single query ($s=0.05$) or distributed across all queries ($s=1.0$).
>
> Since the sensitivity of the agent-level test is driven by the total signal, your suggested experiment hints at the necessity to have a sufficiently large M when designing a query set. We plan to include an extensive version of this result in the appendix.
>
> > 2.
>
> Please see our response to pLsC (1. Calibration Gap).
>
> > 3.
>
> We ran our analysis using 3 additional open-sourced embedding models and 2 additional retrieval depths and compared the ranks of the magnitudes of the group-level test statistics to the rank of the magnitude of the group-level test statistics for (nomic-embed-v1.5, retrieval depth=2). We report the correlation matrices for both experiments.
>
> First, here are the embedding models we considered. All four are open-source and run locally. Importantly, they span different training pipelines (contrastive, distillation), architectures, and embedding dimensions:
>
> |Model|Dimension|Source|
> |---|---|---|
> |nomic-embed-text-v1.5|768|Nomic AI|
> |all-MiniLM-L6-v2|384|Sentence-Transformers|
> |gte-large|1024|Alibaba DAMO (thenlper)|
> |bge-large-en-v1.5|1024|BAAI|
>
> ## Pairwise Spearman rank correlations (embedding models; retrieval depth=2)
>
> ||nomic|minilm|gte|bge|
> |---|---:|---:|---:|---:|
> |nomic|1.000|0.996|0.998|0.992|
> |minilm||1.000|0.997|0.995|
> |gte|||1.000|0.997|
> |bge||||1.000|
>
> ## Pairwise Spearman rank correlations (retrieval depth; embedding model=nomic)
>
> |$n_{\text{tweets}}$|1|2|3|
> |---|---:|---:|---:|
> |1|1.000|0.812|0.831|
> |2||1.000|0.874|
> |3|||1.000|
>
> Both results suggest that our analysis is relatively robust to choice of embedding function and choice of retrieval depth. Space permitting, we will include the sensitivity correlation plots in the main text and include full replications of the group-level experiment in the appendix.

---

> > ### Author Rebuttal · Reviewer_uP9F · 2026-04-03
> >
> > Thank the authors for the detailed response! My concerns have been adequately addressed. I believe this is an valuable idea and would like to slightly raise my score.

---

### Official Review · Reviewer_pLsC · 2026-03-17

**Soundness:** 3
**Presentation:** 3
**Significance:** 3
**Originality:** 3
**Overall Recommendation:** 4
**Confidence:** 4

**Summary:**

This paper extends prior DKPS-style response-based black-box representations to the temporal multi-agent setting. The main idea is TDKPS, which jointly embeds (agent,time) instances into a shared space using distances derived from query-conditioned response embeddings, and then uses this representation to test whether a single agent or a group of agents has changed over time. On top of TDKPS, the paper proposes an agent-level permutation test and a group-level paired energy-distance test. The evaluation includes controlled simulations and a real-data study on digital congresspersons, where the method identifies stronger public-health-related shifts around the COVID period than on control query sets.

**Compliance With Llm Reviewing Policy:**

Affirmed.

**Key Questions For Authors:**

1. How should readers interpret the agent-level results given the calibration gap relative to the group-level results?
2. The method is clearly query-dependent. Do the authors have more concrete guidance on how to construct an informative query set in a new domain?
3. In the main text, it may help to state more explicitly that the method detects broad black-box behavioral shift, rather than isolating which internal or external factor caused it.

**Limitations:**

The limitations and potential negative societal impact have been discussed in this paper.

**Strengths And Weaknesses:**

### Strengths

1. The paper studies a clear and relevant problem: detecting temporal behavioral change in black-box agent systems.
2. The contribution is technically complete: representation, hypothesis tests, simulations, and a nontrivial real-data case study are all included.
3. The method is evaluated in a reasonably disciplined way. The simulations vary effect size, number of agents, number of queries, and number of replicates, and report both power and false positive behavior rather than only qualitative trends.
4. The group-level test is a solid part of the paper: it matches the paired temporal setting well and is much cheaper than repeated re-embedding.

### Weaknesses

1. The agent-level test is not very well calibrated in the reported simulations: the null-class rejection rate is often in the 5–10% range, which makes the agent-level results less convincing than the group-level ones.
2. The empirical evidence is still limited relative to the generality of the claim. The real-data section is interesting, but it is still one application setting centered on one exogenous event pattern.
3. The paper detects that observable behavior changed across time, but it does not identify what component changed. In the current abstraction, the shift could come from the model, tools, retrieved context, or environment, so some of the empirical interpretation would benefit from slightly more careful wording.

---

> ### Author Rebuttal · Authors · 2026-03-27
>
> Thank you for taking the time to provide a detailed review. We think your summary provides a fair representation of our work. We also appreciate your agreement that the problem we study is relevant and that our presentation of our work is clear and that our experimental design is well thought out. We address your questions by number / short description.
>
> > 1. Calibration gap
>
> In our real-data analysis the agent-level calibration gap is not a major concern because we provide highly-paired control conditions and, as such, an inflation in size should affect all conditions equally. Further, we provide complementary analysis via the group-level test that is both faster and calibrated.
>
> With that said, the calibration of the agent-level test was an issue raised by the majority of the reviewers. We designed a more calibrated test based on TDKPS that splits the $ R $ observations into two sets of size $\lfloor R / 2 \rfloor$ and uses one set to estimate the embedding basis (via SVD of the kernel matrix) and the other to compute the test statistic and its permutation null distribution. This test is approximately valid at small $R$ and maintains high power for large $R$. The table below contains the probability of rejection for a system with $N = 25$, $M = 10$, $p = 50$, $p_s = 5$, and $\tau = 0$ (top) and $\tau = 0.5$ (bottom). The probability of rejection for the non-calibrated test is provided for reference. Values are estimated from 100 Monte Carlo replicates with 200 permutations per test and $\alpha = 0.05$; $\pm$ denotes standard error.
> The probability of rejection for the non-calibrated test is provided for reference.
>
> ## $\tau = 0$ (Type 1 Error for both classes)
>
> | $R$ | Class 0 Uncalibrated | Class 0 Calibrated | Class 1 Uncalibrated | Class 1 Calibrated |
> |-----|---------------------:|-------------------:|---------------------:|-------------------:|
> | 10  | 0.072 ± 0.008       | 0.040 ± 0.006     | 0.098 ± 0.009       | 0.053 ± 0.006     |
> | 20  | 0.069 ± 0.008       | 0.047 ± 0.006     | 0.070 ± 0.007       | 0.048 ± 0.006     |
> | 50  | 0.070 ± 0.007       | 0.048 ± 0.007     | 0.072 ± 0.008       | 0.056 ± 0.006     |
>
> ## $\tau = 0.5$ (Power for class 0 / Type 1 Error for class 1)
>
> | $R$ | Class 0 Uncalibrated | Class 0 Calibrated | Class 1 Uncalibrated | Class 1 Calibrated |
> |-----|---------------------:|-------------------:|---------------------:|-------------------:|
> | 10  | 0.828 ± 0.026       | 0.683 ± 0.033     | 0.102 ± 0.009       | 0.039 ± 0.005     |
> | 20  | 0.905 ± 0.017       | 0.780 ± 0.029     | 0.073 ± 0.007       | 0.047 ± 0.006     |
> | 50  | 0.971 ± 0.009       | 0.936 ± 0.014     | 0.071 ± 0.008       | 0.052 ± 0.006     |
>
> Of note, the calibrated test still dramatically outperforms DCorr (not included in table for brevity). We plan to include a description of the calibrated test in the methods section and add it to the agent-level simulation figure (Figure 3).
>
>
> > 2. Guidance for constructing a good query set
>
> As our real data analysis suggests, the query set is a first-class citizen when designing monitoring systems for multi-agent systems in the black-box setting. We do not have concrete guidance in its selection – though in general, the query set should cover the regions of the query space where the operator is most keen to monitor. For example, do not use a query set related to chocolate and candy if you want to monitor agent behavior on public health. Further, given the expense of generating responses for all N agents in a system, we recommend choosing a moderately sized number of queries (too small and the representations are sensitive to very small changes for a given query, too large and the monitoring system is too expensive).
>
> > 3. Black-box detection
>
> Thank you for the suggestion. The method is indeed sensitive to generic changes in the generation pipeline of the blackbox. We will re-frame parts of the results and discussion to emphasize this. We will also improve the description of which causal claims / localizing the change may be appropriate (such as highly controlled environments where the base model is fixed or in the presence of control experiments, etc). We will also emphasize in the discussion that the query set choices, coupled with the natural experiment set up, permit identification of aggregate causal effects.
>
> Thank you again for taking the time to review our work. Please let us know if you have any other questions / comments / concerns that we can address.

---

### Decision · Program_Chairs · 2026-04-30

**Decision:**

Accept (regular)

**Comment:**

Reviewers agreed that this paper addresses an important problem: detecting behavioral changes (“perspective shifts”) in black‑box multi‑agent systems. The paper is sound and well presented. Concerns were initially raised about aspects of the experimental setup (e.g. calibration of the agent‑level test and soundness of the empirical results), but these were satisfactorily addressed during the rebuttal. Although some reviewers also raised broader questions like how to handle agents with internal state or memory or how to systematically construct the query set, the authors acknowledged these limitations and committed to explicitly discussing them as future work, which I believe is reasonable.

Overall, this work is sound, tackles an important problem, and makes a contribution that is likely to be of interest to the community.